# From Pseudo-Balancing to True Specialization: Memory-Aware Routing for Mixture-of-Experts

## Abstract

Mixture-of-Experts(MoE) efficiently trains large models by using sparse activation to lower costs, selecting a few experts based on data characteristics. For MoE, an unbalanced expert load will lead to routing collapse or increased computational overhead. Existing methods commonly achieve an expert-centered balancing strategy to solve it, prioritizing equal utilization of experts over semantic alignment between tokens and experts. However, this can lead to a pseudo-balance phenomenon: To ensure expert load balancing, the same input is randomly routed to different experts across training steps instead of the most matching one. It introduces two critical issues: (1) Severe knowledge overlap among experts, resulting in redundant representations and inefficient parameter utilization. (2) Difficulty in forming and stabilizing expert specialization. These issues limit the scalability of models, especially large language models(LLM). To address these limitations, we introduce Memory-Aware Routing (MAR), a training-phase approach that enhances existing load-balancing strategies. By equipping each expert with a memory buffer, our method explicitly models their long-term preferences, allowing historical experience to guide routing. This ensures that tokens are routed more consistently to compatible experts, mitigating the pseudo-balance problem while maintaining global load balance and fostering expert specialization. Experimental results show that Memory-Aware Routing improves expert specialization by 35% and downstream accuracy by 2%-25%, doubles parameter efficiency, and matches baseline performance with only half the experts (one-quarter of the parameters).[1]

## 1 Introduction

In the domain of large-scale deep learning, Mixture-of-Experts (MoE) models have emerged as a powerful paradigm (Jacobs et al., 1991b; Roller et al., 2021; Zhou et al., 2022; Jordan & Jacobs, 1994b; Lepikhin et al., 2020). By leveraging a gating mechanism to dynamically route computations to a subset of expert subnetworks (Fedus et al., 2022; Jiang et al., 2024), MoE successfully scales model capacity while maintaining a manageable computational cost (Dai et al., 2024; Shen et al., 2024; Wei et al., 2024; Jiang et al., 2024). However, in the absence of explicit constraints, MoE training often suffers from severe load imbalance, where a few experts are overly activated while others remain underutilized (Lepikhin et al., 2020; Fedus et al., 2022; Zoph et al., 2022; Qiu et al., 2024). This imbalance leads to inefficient resource usage and hinders effective training of all experts.

To mitigate this issue, a variety of expert-centered balancing strategies have been proposed. GShard (Lepikhin et al., 2020) improved upon earlier work (Shazeer et al., 2017b) by introducing the differentiable auxiliary loss $L_{\text{aux}}$, which has since been widely adopted (Fedus et al., 2022; Jiang et al., 2024; Lieber et al., 2024; Mosaic Research Team, 2024). On larger scales, Zoph et al. (2022) introduced the z-loss $L_{\text{z}}$ to enhance training stability. Related methods (Shazeer et al., 2017a; Dai et al., 2024; Qiu et al., 2025) advanced load-balancing with soft constraints, dynamic reallocation, and multi-loss routing, while some explored auxiliary-loss-free strategies (DeepSeek-AI, 2024).

Although expert-centered load-balancing methods effectively equalize expert utilization, they overlook semantic alignment between tokens and experts during routing. This oversight leads to the

---

[1]Our code is available at `https://anonymous.4open.science/r/MAR-MoE-F7D1/`

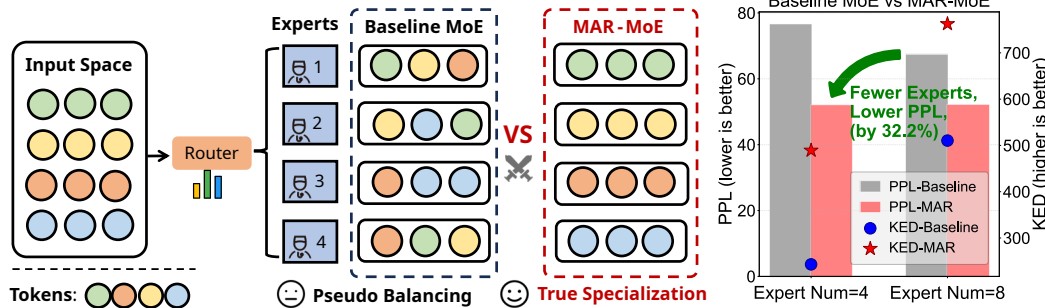

Figure 1: Pseudo-Balancing Vs. True Specialization And Results. Left: Pseudo-balancing equalizes load but misaligns semantics, causing expert overlap and impeding specialization; true specialization co-routes similar tokens, yielding distinct expertise and less redundancy. Right: MAR-MoE augments load-balanced Llama-MoE (baseline MoE) with Memory-Aware Routing, achieving 50% fewer experts (25% fewer parameters) without loss and a 35% increase in specialization (KED).

emergence of the **pseudo-balance phenomenon**. In practice, to maintain balanced usage, identical tokens may be randomly routed to different experts across training steps, preventing their stable assignment to the most semantically appropriate expert. Figure 1 provides a high-level illustration of this phenomenon, and we formally define it below.

Let $\mathcal{X}$ denote the input data space, and $\mathcal{E} = \{E_1, E_2, \ldots, E_N\}$ the set of $N$ experts. A standard MoE model employs a routing function $R(x)$ that assigns an input token $x \in \mathcal{X}$ to one or more experts. Formally, we define a pseudo-balanced routing function $R_{\text{pseudo}}(x, t)$ which, for a token $x$ at the training step $t$, assigns it to an expert $E_i \in \mathcal{E}$ primarily according to load-balancing constraints rather than token content. Let $p(E_i \mid x, t)$ denote the probability of routing token $x$ to expert $E_i$ at step $t$. Under pseudo-balance, this probability satisfies

$$p(E_i \mid x, t) \approx \frac{1}{N}, \quad \forall E_i \in \mathcal{E}, \ \forall x \in \mathcal{X}.$$

As a result, the mapping between samples and experts remains unstable, preventing experts from accumulating experience on consistent data distributions. This instability causes expert learning to resemble enforced sharing rather than dedicated specialization, which in turn leads to substantial functional overlap across experts and hinders the development of true specialization. Such redundancy not only wastes parameters but also ultimately constrains the scalability of MoE models.

The core challenge in addressing this issue lies in the inherent trade-off between expert-centered load balancing and expert specialization. Enforcing strict balancing introduces excessive randomness that undermines specialization, whereas relaxing the balancing constraint risks expert collapse.

To address these limitations, we propose Memory-Aware Routing (MAR), a novel mechanism that augments load balancing with memory-guided routing. The MAR introduces memory buffers to explicitly capture the long-term preferences of each expert, guiding the model to avoid assigning similar information to different experts and effectively mitigating knowledge overlap. In addition, we define an expert–token matching score, which quantifies the similarity between an input token and an expert's preference vector. This score promotes consistent routing of tokens to semantically aligned experts, fostering the emergence and consolidation of expert specialization. By maintaining global balance while transitioning from uniform allocation to differentiated routing, MAR mitigates the pseudo-balance problem and encourages both functional diversity and stable specialization across experts. The significant performance improvements achieved by MAR are highlighted in Figure 1. Notably, MAR applied only during training without incurring extra overhead at inference time.

In summary, our main contributions are as follows:

- We first reveal the pseudo-balance phenomenon, where load-balancing mechanisms reassign the same input to different experts across training steps, hindering specialization and wasting parameters, and ultimately limiting model scalability.

- We propose Memory-Aware Routing, where expert-specific memory buffers model long-term preferences to guide routing, ensuring consistent token-to-expert assignment, mitigating pseudo-balancing, and promoting functional diversity and stable specialization.

- We conduct extensive evaluations. The experiments illustrate our method reduces the number of experts by 50% and the total parameters by one quarter without sacrificing performance, resulting in lower computational cost, and it enhances expert specialization by 35% and yields 2%–25% accuracy gains across multiple tasks.

## 2 RELATED WORK

### 2.1 MIXTURE OF EXPERT

Mixture-of-Experts (MoE) traces back to the statistical learning literature, where Jacobs et al. (1991a); Jordan & Jacobs (1994a) introduced (hierarchical) mixtures trained with EM to encourage specialization among subnetworks. In modern deep learning, Shazeer et al. (2017c) revived MoE at scale with sparsely-gated layers that route each token to a small subset of experts, achieving large capacity without proportional compute. Building on this, Du et al. (2022) demonstrated trillion-parameter MoE language models with strong few-shot performance at lower training energy than dense baselines. Recent open models also validate MoE's quality/efficiency in practice. Mixtral 8×7B Jiang et al. (2024) attains strong results with two-expert routing per token; DBRX employs fine-grained MoE with smaller experts to improve Pareto efficiency Mosaic Research Team (2024). DeepSeek-V2 DeepSeek-AI et al. (2024) combines architectural changes (e.g., MLA) with fine-grained MoE to reduce active parameters and KV cache costs while maintaining performance.

### 2.2 LOAD BALANCING

Unconstrained gating in MoE tends to overuse a few experts, leaving others idle. To counter this, Shazeer et al. (2017b) introduced the importance loss $L_{\text{importance}}$ and load loss $L_{\text{load}}$, which Lepikhin et al. (2020) distilled into a differentiable auxiliary loss $L_{\text{aux}}$. Its effectiveness has been validated by Fedus et al. (2022) and widely adopted in practice (Jiang et al., 2024; Lieber et al., 2024; Mosaic Research Team, 2024). Zoph et al. (2022) identified limitations at larger scales, prompting the introduction of the $z$-loss $L_{\text{z}}$ to improve training stability. In related efforts, Shazeer et al. (2017a) applied soft constraints with auxiliary losses and stochastic smoothing for differentiable load evaluation; Dai et al. (2024) improved utilization via load-balancing losses, dynamic reallocation, Residual-MoE, and cross-GPU parallelism; Qiu et al. (2025) introduced multiple auxiliary losses to build balanced routing; and DeepSeek-AI (2024) explored an auxiliary-loss-free strategy with expert-specific biases. While these approaches mitigate imbalance, they still suffer from the pseudo-balance problem, where identical inputs are inconsistently routed across steps, leading to unstable mappings, knowledge redundancy, and limited specialization.

## 3 THE PSEUDO-BALANCE PHENOMENON

In Mixture-of-Experts (MoE) training, load-balancing strategies are commonly employed to prevent overuse of certain experts while leaving others underutilized. However, these strategies often give rise to the pseudo-balance phenomenon, wherein identical inputs are forced to be routed to different experts rather than consistently assigned to the most semantically aligned ones. This effect is inherent to the load-balancing loss itself:

$$\mathcal{L}_{\text{balance}} = \alpha \sum_{i=1}^{N} \text{Load}(i)\, p_i,$$

where $\text{Load}(i)$ denotes the proportion of tokens assigned to expert $i$ in the current batch, and $p_i$ represents the average gating probability allocated to expert $i$. The gradients of this loss, $\partial \mathcal{L}/\partial p_i = \alpha f_i$ and $\partial \mathcal{L}/\partial f_i = \alpha p_i$, create a symmetric feedback mechanism that penalizes any expert receiving more tokens or having a higher routing probability. The loss is minimized at the uniform fixed point $f_i = p_i = 1/E$, analytically enforcing equal usage across experts. To rigorously examine the resulting pseudo-balance phenomenon, we design experiments from two complementary perspectives: training-time expert selection and post-training expert specialization.

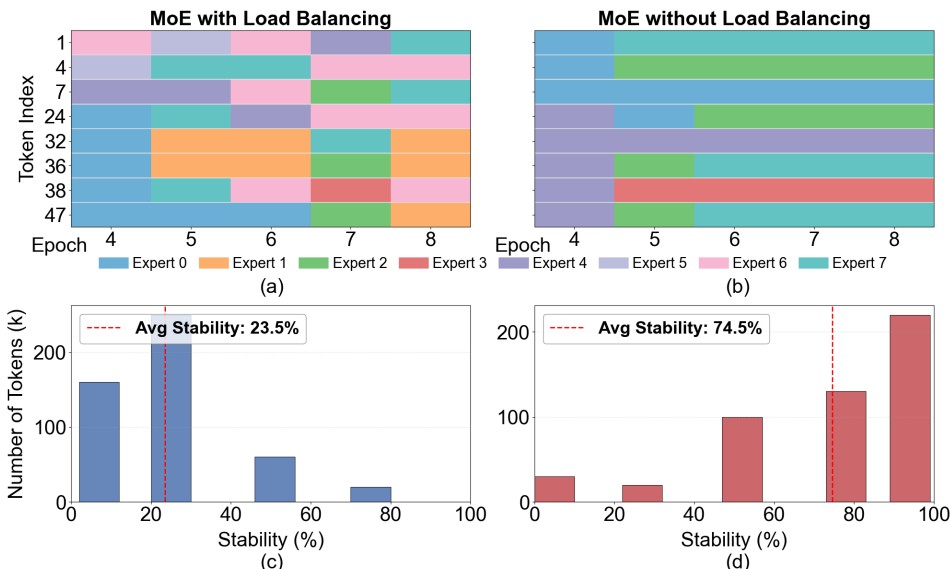

Figure 2: Experimental Validation of the Pseudo-Balance Phenomenon. Subfigures (a) and (b) show expert assignments with and without load balancing: the balanced model keeps switching experts, while the baseline stabilizes earlier. Subfigures (c) and (d) measure assignment consistency across epochs, showing a 68% drop under load balancing.

## 3.1 EXPERT SELECTION DYNAMICS DURING TRAINING

To understand how load balancing impacts the routing process, we first analyzed the stability of expert assignments throughout the training. Our hypothesis was that a lack of stable assignments would prevent experts from specializing. We compared a model using a load-balancing strategy with a baseline model without one. As shown in Figure 2 (a) and (b), even in later stages of training, when routing is expected to have converged, the model with load balancing still shows frequent and random switching of the same inputs across different experts. In stark contrast, the baseline model achieves a stable and consistent assignment for a given input much earlier in training. The experimental setup details are provided in the Appendix A.1.1.

For a quantitative assessment of this instability, we measured the consistency of expert assignments across different epochs for the same input samples. Our results, illustrated in Figure 2 (c) and (d), reveal a dramatic 68% drop in assignment consistency for the load-balanced model compared to the baseline. This shows that expert-centered load balancing methods significantly undermine the determinism of routing decisions, preventing experts from accumulating consistent experience and thereby hindering the emergence of stable specialization.

## 3.2 POST-TRAINING ANALYSIS OF EXPERT SPECIALIZATION

The routing instability we observed during training has a direct impact on the model's final state: a high degree of knowledge redundancy among experts. Since multiple experts are repeatedly forced to handle the same type of input, their learned functional representations converge, leading to significant overlap and reduced parameter efficiency.

To validate this redundancy, we adopt the "expert disabled" experiment proposed by Dai et al. (2024). In this test, we progressively disabled a varying proportion of top-rank experts and measured the resulting change in perplexity (PPL). The core idea is that if the model performance remains largely unaffected after disabling experts, it implies high functional redundancy. Conversely, a steep drop in performance indicates clearer specialization, as removing a unique expert's contribution would have a greater impact.

As depicted in Appendix A.2.1, the model with load balancing exhibits far less sensitivity to expert masking. The perplexity remains relatively low even when a significant portion of its top experts are disabled, demonstrating a high degree of redundancy. In contrast, the baseline model without

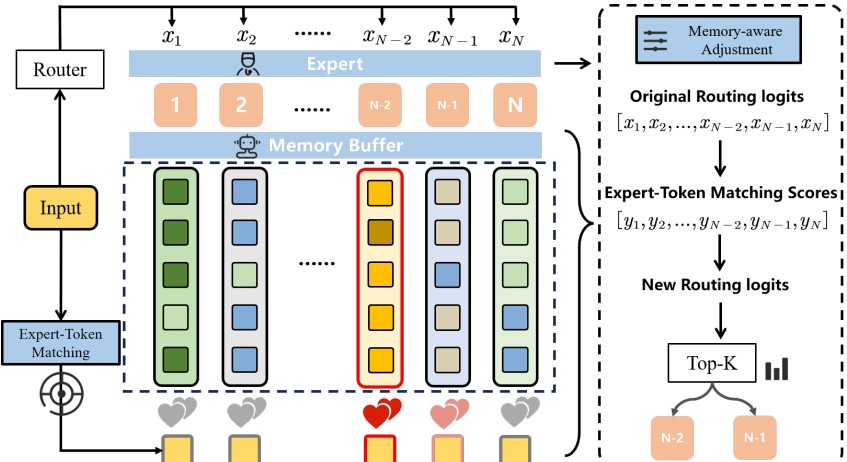

Figure 3: Overall framework of MAR. Unlike vanilla MoE, which relies solely on router logits and Top-k selection, MAR introduces an expert–token matching step to better align tokens with suitable experts during training. Each expert maintains a memory buffer that derives a preference vector capturing long-term tendencies. Token representations are compared against these vectors, and the resulting matching scores are fused with the original logits through an interest-aware adjustment to form new routing logits. Final expert selection is determined by Top-k. This mechanism ensures consistent and semantically aligned token assignments while preserving global balance.

load balancing shows a much steeper performance degradation, confirming that its experts have successfully specialized and are not interchangeable. These results provide direct evidence that balancing methods cause substantial knowledge overlap among experts.

Our findings indicate that although expert-centered balancing strategies achieve a superficial global balance, they create a pseudo-balance phenomenon, which arises from the increased randomness in routing. By forcing identical inputs to be assigned to different experts, this effect gives rise to two critical problems: a high degree of knowledge overlap and a failure to achieve stable specialization. Consequently, the core advantages of MoE—parameter efficiency through specialization—are diminished, fundamentally restricting the model's scalability.

## 4 MEMORY-AWARE ROUTING: TOWARDS TRUE SPECIALIZATION

In essence, the central question is how we can transition from pseudo-balance to true expert specialization without losing the benefits of load balancing? To tackle this challenge, we introduce a novel approach: Memory-Aware Routing (MAR). The overall framework of Memory-Aware Routing is illustrated in Figure 3. To address severe knowledge overlap among experts, MAR introduces a memory buffer that stores representations of recently processed tokens. By aggregating these historical representations, each expert can maintain a long-term preference vector, which guides routing decisions and prevents multiple experts from redundantly learning the same information. To form and consolidate expert specialization, we propose the expert–token matching score, defined as the similarity between an incoming token and an expert's preference vector. During routing, this score is combined with the original routing score through weighted fusion, encouraging input to be consistently assigned to semantically aligned experts, therefore promoting expert specialization.

### 4.1 MEMORY BUFFER

In our framework, each expert is equipped with an independent memory buffer that stores the representations of tokens it has recently processed. Let the number of experts be $K$ and the hidden dimension be $d$. The memory buffer of the $i$-th expert can be expressed as

$$\mathcal{B}_i = \{h_i^{(1)}, h_i^{(2)}, \ldots, h_i^{(N)}\}, \quad i = 1, \ldots, K,$$

where $h_i^{(j)} \in \mathbb{R}^d$ denotes the representation of a token routed to the expert $i$, and $N$ is the buffer capacity. To maintain its representational freshness, the buffer is updated with a FIFO strategy: Each newly routed token is appended, and once the buffer is full, the earliest entry is discarded. This

mechanism ensures that the stored vectors consistently reflect the recent routing history of the expert, while preventing outdated information from dominating. Also, FIFO is a straightforward queue operation with negligible computational cost, enabling the model to update its memory efficiently during training without becoming a performance bottleneck.

By aggregating the buffered feature vectors, we obtain a preference vector for expert $i$:

$$d_i = \frac{1}{|\mathcal{B}_i|} \sum_{h \in \mathcal{B}_i} h, \quad d_i \in \mathbb{R}^d.$$

This preference vector is dynamically updated during training, effectively recording the types of information the expert has processed and guiding future routing decisions. This ensures each expert can specialize in the tasks its most proficient at, fundamentally reducing knowledge overlap among experts and enhancing the overall specialization and performance of the entire multi-expert system.

### 4.2 EXPERT-TOKEN MATCHING SCORE

To promote expert specialization, we propose the Expert-Token Matching Score, which explicitly measures the compatibility between each token and the long-term preferences of experts. Given an input token representation $x \in \mathbb{R}^d$, the gating network first computes a base score for each expert:

$$s_i^{\text{base}} = \text{Router}(x)_i, \quad i = 1, \ldots, K,$$

To encourage experts to develop differentiated functional preferences, we define the Expert-Token score as the cosine similarity between the token and the expert's preference vector $d_i$:

$$s_i^{\text{match}} = \cos(x, d_i) = \frac{x \cdot d_i}{\|x\| \|d_i\|}.$$

If $d_i$ is a zero vector, the similarity is defined as 0. The final routing score is then obtained by combining the two components through weighted fusion:

$$s_i = s_i^{\text{base}} + \alpha \cdot s_i^{\text{match}},$$

where the hyperparameter $\alpha \in [0, 1]$ controls the relative influence of long-term preferences on the routing decision. Based on the final scores $s_i$, the model selects the top-$k$ experts for forward computation, and the corresponding token representations are subsequently written into their memory buffers to update the expert preference vectors. In this way, routing decisions no longer rely solely on instantaneous token features but also incorporate historical preferences, ensuring that tokens are directed to semantically aligned and specialized experts. This design effectively alleviates the pseudo-balance phenomenon and fosters functional diversity and stable expert specialization.

### 4.3 COMPLEXITY AND EFFICIENCY

MAR is applied only during training and does not introduce additional trainable parameters. Updating each expert's preference vector aggregates features from its buffer with complexity $\mathcal{O}(Nd)$ ($N$: buffer size, $d$: hidden dimension), and computing token–expert similarities during routing costs $\mathcal{O}(Kd)$ per token ($K$: number of experts), preserving scalability without introducing training bottlenecks. Moreover, a quantitative comparison of peak GPU memory usage and routing latency between LBL and LBL+MAR (reported in Appendix A.2.4) indicates that the additional training-time overhead introduced by MAR is minimal. At inference, expert–token matching is disabled and preference vectors are not used, so routing reverts to standard top-$k$ selection based on gating logits with identical FLOPs, memory and latency as a vanilla MoE. Crucially, smooth training-time updates of the preference vectors stabilize routing and reduce token oscillation across experts, fostering consistent specialization.

## 5 EXPERIMENTS

### 5.1 EXPERIMENTAL SETUPS

#### 5.1.1 MODEL ARCHITECTURE AND TRAINING SETTINGS

We conduct experiments on four representative MoE architectures: Mixtral-MoE (Jiang et al., 2024), LLaMA-MoE (Zhu et al., 2024), GPT2-MoE (Lagler et al., 2013), and OLMoE (Muennighoff et al.,

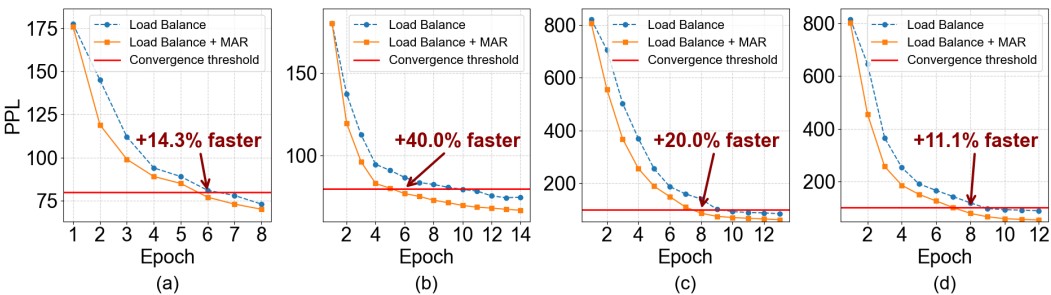

Figure 4: Perplexity convergence of Mixtral-MoE and Mixtral-MoE+MAR during training. Experiments are conducted on PTB with 4 experts (a) and 8 experts (b), and on WikiText-2 with 4 experts (c) and 8 experts (d). Models with MAR achieve faster PPL reduction within the same number of training steps and reach stable performance in fewer iterations, demonstrating that MAR lowers training cost and accelerates convergence.

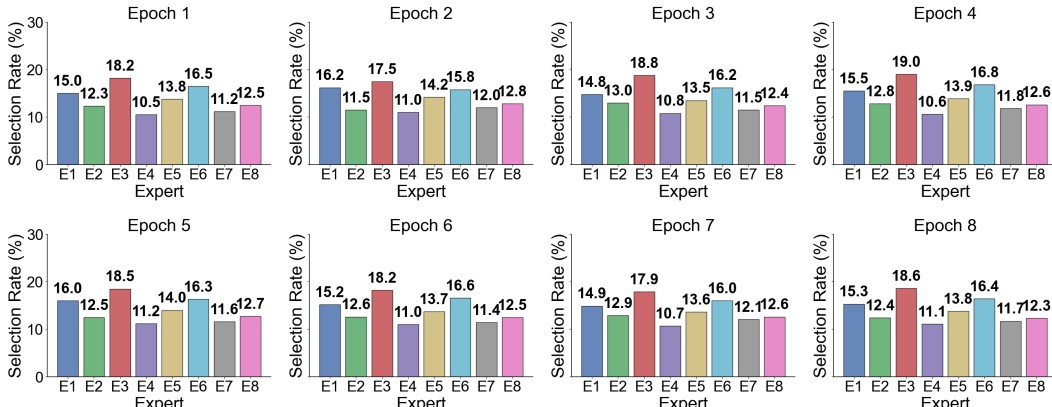

Figure 5: Expert load distribution across training epochs in the Mixtral-MoE+MAR model. Models with MAR exhibit stable and balanced utilization of experts, indicating the effectiveness of Memory-Aware Routing in mitigating expert collapse.

2024). For GPT2-MoE, all FFN layers in Transformers (Vaswani et al., 2017) are replaced with MoE layers. To comprehensively evaluate the effectiveness of the Memory-Aware Routing,we conduct experiments from two perspectives: the pretrain phase and the fine-tuning phase. In the pretrain phase, we evaluate whether MAR could overcome parameter redundancy caused by the expert redundancy. In the fine-tuning phase, we further verify the performance of the downstream tasks.

Unless noted, all MoE models with load balancing follow the predominant entropy-based load-balancing strategy (Lepikhin et al., 2020; Fedus et al., 2022). Each MoE layer has 8 experts (hidden size 512) with top-2 routing (Jiang et al., 2024); the load-balancing loss weight is 0.4. In MAR models, each expert keeps a memory buffer of 128 with a memory coefficient 0.5. We use GPT2Tokenizer (max length 2048), AdamW (Loshchilov & Hutter, 2017) with gradient accumulation and a linear schedule, and early stopping on validation loss. Experiments run on 5×RTX 4090 and 2×L20 GPUs. Full architectural and hyperparameter details are in the Appendix A.1.

For the pre-training stage, we adopt different datasets for different models. Mixtral-MoE, LLaMA-MoE and GPT2-MoE (with 3 MoE layers) are trained on PTB (Marcinkiewicz, 1994), and Wikitext-2 (Merity et al., 2016), while GPT2-MoE (with 12 MoE layers) is pre-trained on OpenWebText (Gokaslan et al., 2019). For fine-tuning, we use the OLMoE 1B-7B model (Muennighoff et al., 2024) and evaluate it on a suite of downstream datasets covering language modeling, commonsense reasoning, and mathematical reasoning.

### 5.1.2 EVALUATION

Table 1: Perplexity (PPL) and Key Expert Dependency (KED) of Mixtral-MoE, Llama-MoE, and GPT2-MoE with 3 MoE layers across different datasets and expert numbers. Memory-Aware Routing (MAR) consistently reduces PPL and improves KED compared to vanilla load balancing loss (LBL), highlighting both parameter efficiency and scalability.

| Model | PTB dataset | | | | WikiText-2 dataset | | | |
|---|---|---|---|---|---|---|---|---|
| | Num=4 | | Num=8 | | Num=4 | | Num=8 | |
| | PPL($\downarrow$) | KED($\uparrow$) | PPL($\downarrow$) | KED($\uparrow$) | PPL($\downarrow$) | KED($\uparrow$) | PPL($\downarrow$) | KED($\uparrow$) |
| **Mixtral-MoE Model** | | | | | | | | |
| LBL | 74.48 | 105.32 | 72.33 | 129.24 | 81.37 | 91.89 | 78.25 | 137.64 |
| LBL+MAR | **69.68** | **152.95** | **69.43** | **186.72** | **72.96** | **143.31** | **72.18** | **172.30** |
| Gain (%) | -6.37% | +45.11% | -4.03% | +44.48% | -10.33% | +55.98% | -7.78% | +25.17% |
| **Llama-MoE Model** | | | | | | | | |
| LBL | 59.34 | 103.86 | 56.88 | 157.31 | 76.50 | 242.97 | 67.38 | 510.45 |
| LBL+MAR | **45.18** | **168.08** | **44.81** | **205.29** | **51.86** | **289.20** | **50.97** | **662.53** |
| Gain (%) | -23.85% | +61.82% | -21.23% | +30.54% | -32.20% | +19.03% | -24.35% | +29.81% |
| **GPT2-MoE Model** | | | | | | | | |
| LBL | 62.10 | 98.45 | 58.75 | 145.60 | 80.25 | 210.33 | 70.10 | 450.12 |
| LBL+MAR | **54.88** | **140.72** | **51.23** | **195.87** | **68.90** | **265.44** | **62.15** | **620.50** |
| Gain (%) | -11.63% | +42.90% | -12.81% | +34.59% | -14.14% | +26.17% | -11.29% | +37.87% |

We assess our method using perplexity $P$ for pre-training and task-specific metrics (e.g., accuracy) for downstream tasks. We further introduce *Key Expert Dependency* (KED) to quantify expert specialization. Following Dai et al. (2024), models with lower expert redundancy are more sensitive to disabling their most-routed experts. Let $N$ denote the total number of experts and $n$ the minimum number of experts that must remain active during routing (e.g., $n = 2$ for top-2). For each $k = 1, \ldots, N-n$, we sequentially disable the $k$ top-routed experts and record the resulting perplexity $P(k)$, with $P(0)$ denoting the original perplexity. We then define

$$\text{KED} = \frac{1}{N-n} \sum_{k=1}^{N-n} \frac{P(k) - P(0)}{k}.$$

Higher KED indicates stronger reliance on a small subset of experts, whereas lower KED suggests more complementary specialization across experts. Baselines use standard load-balancing routing, while our method applies MAR. All runs use identical hyperparameters and optimization.

## 5.2 Main Results

### 5.2.1 Pre-training Stage

On the base models Mixtral-MoE, LLaMA-MoE and GPT2-MoE (with 3 MoE layers), we conduct comparative experiments with and without MAR. We record the expert selections for each training epoch. As shown in Appendix A.2.2, in models with standard load balancing, the same inputs continue to oscillate between different experts even in the later stages of training, whereas models with MAR exhibit stable expert assignments. This indicates that MAR effectively alleviates the pseudo-balance phenomenon during training. See Appendix A.1.2 for experimental settings.

We further compare convergence speed between the two settings. As illustrated in Figure 4, our MAR models achieve an average PPL reduction of 5% to 20% greater than the baseline model over the same number of training steps. It also reached stable performance in approximately 20% fewer iterations. This demonstrates that MAR lowers training cost and accelerates convergence.

We additionally investigated the effectiveness of MAR in preserving expert load balance during training. To this end, we visualize the distribution of expert load across epochs, as shown in Figure 5. The results demonstrate MAR effectively supports a balanced use of experts, avoiding the collapse observed in the absence of load balancing methods. More detalis are presented in Appendix A.2.3.

Table 2: Performance of applying MAR on top of different load-balancing strategies. 'Global Batch LBL' refers to Qiu et al. (2025). 'Aux Free' refers to Wang et al. (2024). 'Lo+Lv' refers to Guo et al. (2025). 'SimBal' refers to Omi et al. (2025). MAR consistently reduces PPL and improves KED compared to each original load-balancing loss. Gains indicate improvement of MAR over the baseline: PPL decrease or KED increase are shown in red; deterioration would be shown in blue.

| Method | PTB dataset | | | | WikiText-2 dataset | | | |
| | Mixtral | | LLaMA | | Mixtral | | LLaMA | |
| | PPL($\downarrow$) | KED($\uparrow$) | PPL($\downarrow$) | KED($\uparrow$) | PPL($\downarrow$) | KED($\uparrow$) | PPL($\downarrow$) | KED($\uparrow$) |
|---|---|---|---|---|---|---|---|---|
| Global Batch LBL | 70.42 | 160.51 | 42.25 | 188.37 | 73.34 | 168.32 | 55.03 | 535.21 |
| + MAR (Ours) | **67.79** | **217.05** | **40.74** | **236.91** | **70.44** | **192.23** | **49.66** | **688.33** |
| Gain (%) | -3.63% | +56.54% | -1.51% | +48.54% | -2.90% | +23.91% | -5.37% | +153.12% |
| Aux Free | 70.83 | 154.03 | 42.21 | 189.41 | 73.56 | 166.73 | 54.06 | 601.54 |
| + MAR (Ours) | **67.52** | **225.98** | **41.28** | **243.24** | **70.01** | **198.24** | **50.56** | **669.29** |
| Gain (%) | -3.31% | +71.95% | -0.93% | +53.83% | -3.55% | +31.51% | -3.50% | +67.75% |
| Lo+Lv | 71.95 | 141.48 | 42.11 | 167.55 | 76.12 | 149.13 | 54.44 | 595.21 |
| + MAR (Ours) | **68.98** | **198.67** | **41.77** | **214.43** | **71.97** | **183.66** | **50.07** | **674.86** |
| Gain (%) | -2.97% | +57.19% | -0.34% | +46.88% | -4.15% | +34.53% | -4.37% | +79.65% |
| SimBal | 72.30 | 139.59 | 42.02 | 176.87 | 77.07 | 145.75 | 54.31 | 595.95 |
| + MAR (Ours) | **69.01** | **191.88** | **40.54** | **221.92** | **71.24** | **188.50** | **50.03** | **679.04** |
| Gain (%) | -3.29% | +52.29% | -1.48% | +45.05% | -5.83% | +42.75% | -4.28% | +83.09% |
| Shared Experts | 71.64 | 159.29 | 36.72 | 235.24 | 73.98 | 172.64 | 40.32 | 710.32 |
| + MAR (Ours) | **69.32** | **205.31** | **36.25** | **254.57** | **69.96** | **212.39** | **39.50** | **714.82** |
| Gain (%) | -2.32% | +46.02% | -0.47% | +19.33% | -4.02% | +39.75% | -0.82% | +4.50% |

Table 3: Perplexity (PPL) and Key Expert Dependency (KED) of GPT2-MoE with 12 MoE layers using 8 and 16 experts. Incorporating MAR on top of the load balance loss (LBL) consistently reduces PPL by 8.6%–9.4% and improves KED by 31.2%–47.1% compared to the vanilla LBL.

| Model | Num=8 | | Num=16 | |
| | PPL($\downarrow$) | KED($\uparrow$) | PPL($\downarrow$) | KED($\uparrow$) |
|---|---|---|---|---|
| GPT2-MoE +LBL | 52.45 | 91.89 | 48.24 | 124.26 |
| GPT2-MoE +LBL+MAR | **47.94** (-8.6%) | **135.14** (+47.1%) | **43.71** (-9.4%) | **163.05** (+31.2%) |

For model performance, we evaluate perplexity (PPL) as the primary metric and Key Expert Dependency (KED) to measure expert specialization. Table 1 shows that MAR substantially improves expert specialization, resulting in an average gain of 35% in KED. In both base models, MAR achieves superior performance while using only half the number of experts (reducing total parameters by 25%). This highlights simultaneous gains in parameter efficiency and training efficiency. Furthermore, we compare MAR with other load-balancing strategies (Qiu et al., 2025; Wang et al., 2024; Guo et al., 2025; Omi et al., 2025), including shared-expert architectures (DeepSeek-AI et al., 2024), and Table 2 demonstrates that MAR provides consistent improvements across all baselines.

To examine scalability, we further apply MAR to GPT2-MoE with 12 MoE layers, whose parameter count is four times that of GPT2-MoE with 3 MoE layers, pre-trained on the OpenWebText dataset (Gokaslan et al., 2019). Table 3 confirm the same trends, suggesting that MAR consistently enhances expert specialization and parameter utilization across different model scales.

### 5.2.2 FINE-TUNING STAGE

For fine-tuning, we initialize with the open-source OLMoE 1B-7B model (Muennighoff et al., 2024) and conduct experiments with and without MAR across four representative downstream tasks: language modeling, knowledge application, commonsense reasoning, and mathematical reasoning. Specifically, language modeling is evaluated on the PTB dataset (Marcinkiewicz, 1994) using perplexity (PPL) as the metric. Knowledge application is measured on MMLU (Hendrycks et al.,

Table 4: Performance comparison of OLMoE 1B-7B with and without Memory-Aware Routing (MAR) on PTB, MMLU, SVAMP, BBH, and GSM8K. With MAR, PPL decreases by 8.9% and accuracy improves by 2.2%–25.4% over load balance, demonstrating consistent and substantial gains across diverse benchmarks.

| Model | PTB | MMLU | SVAMP | BBH | GSM8K |
|---|---|---|---|---|---|
| | PPL($\downarrow$) | Acc($\uparrow$) | Acc($\uparrow$) | Acc($\uparrow$) | Acc($\uparrow$) |
| LBL | 17.02 | 43.06 | 10.60 | 17.12 | 7.66 |
| LBL+MAR | **15.49** (-8.9%) | **44.08** (+2.2%) | **13.30** (+25.4%) | **18.16** (+6.07%) | **8.26** (+7.8%) |

2021), while commonsense reasoning is assessed with BBH (Suzgun et al., 2022). For mathematical reasoning, we adopt GSM8K (Cobbe et al., 2021) and SVAMP (Patel et al., 2021) as evaluation benchmarks. Further details are provided in Appendix A.1.3.

Table 4 demonstrate that models with MAR consistently outperform their counterparts, achieving performance improvements of 2%–25% across all task, further validating the effectiveness of the proposed method in practical applications.

### 5.3 Ablation Study

To further analyze the key design choices of Memory Aware Routing, we conduct ablation experiments from two perspectives: the influence factor of memory $\alpha$ and the memory buffer size. The results, presented in Appendix A.1.4, demonstrate that setting the influence factor of memory $\alpha$ to 0.5 optimally combines current and long-term preferences, effectively breaking pseudo-balance and accelerating expert specialization. However, an excessively high $\alpha$ leads to a decline in performance due to "path dependence" and overfitting to past data. Similarly, an optimal buffer size of 128 is crucial; a buffer that is too small leads to instability, while a buffer that is too large introduces outdated information, ultimately degrading the model's generalization ability. These findings validate that achieving a proper balance is key to efficient expert specialization and stable model convergence.

In addition, we perform a more detailed comparison of different memory–buffer update strategies and alternative formulations of the expert–token matching score (results provided in Appendix A.2.5). The analysis demonstrates that FIFO achieves an optimal balance between effectiveness and computational efficiency.

### 6 Conclusion

In this work, we first reveal a key limitation of expert-centered load balancing in MoE models: the pseudo-balance phenomenon, where the same input is routed to different experts across training steps for global balancing, rather than consistently to the most semantically aligned expert. This leads to high functional overlap among experts and hinders the formation of distinct specialization, thereby limiting the scalability of MoE models. To address this, we propose Memory-Aware Routing (MAR), which augments traditional load balancing with memory-guided routing. MAR ensures that tokens are consistently assigned to the most compatible experts, maintaining balanced usage while improving assignment stability and rationality, effectively mitigating the pseudo-balance issue. MAR increases expert differentiation by 35%, allows halving the number of experts without performance loss (improving parameter utilization by 25%), and achieves 2%–25% performance gains across downstream tasks. Our findings show that effective routing is the key to promoting true expert specialization. A promising area for future research is to explore how to amplify this specialization through improved routing, ultimately leveraging the core benefits of MoE models. Moreover, MAR can facilitate MoE compression and expert pruning by stabilizing token-to-expert assignments and promoting expert differentiation, enabling more efficient model reduction.

### 7 Reproducibility Statement

To support reproducibility, we document architectures, hyperparameters, and training procedures in the main text and Appendix A.1.2; preprocessing steps, evaluation metrics, and additional results are provided in Appendices A.2.3 and A.1.3. An anonymous code and configuration repository is linked in the main text to enable exact replication.

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

## A  APPENDIX

### A.1  IMPLEMENTATION DETAILS

#### A.1.1  DEMONSTRATION OF A PSEUDO-BALANCE PHENOMENON

We build our model upon a lightweight Mixture-of-Experts Transformer that follows the design paradigm of Mixtral-MoE (Jiang et al., 2024). Specifically, the architecture, extends a Transformer backbone by replacing the standard feed-forward sublayer with a Mixture-of-Experts (MoE) module. Each Transformer block first applies multi-head self-attention with a hidden dimension of 1024 and 8 attention heads, followed by the MoE sublayer. The MoE module consists of eight experts, where each expert is implemented as a two-layer feed-forward network of dimension $512 \rightarrow 512 \rightarrow 512$ with GELU activation. A learned router assigns tokens to experts using top-2 gating, and the final output is obtained by weighting the selected experts with softmax-normalized routing scores and combining their outputs. Residual connections and layer normalization are applied after both the attention and MoE sublayers.

In models augmented with load balancing, we incorporate an entropy-based regularization term into our training objective. This term, $\mathcal{L}_{\text{balance}}$, encourages the model to distribute tokens uniformly across all experts. Let $\text{Load}(i)$ denote the empirical load of the $i$-th expert, calculated as the fraction of tokens assigned to it within a batch. The load-balancing loss is then formulated as a scaled entropy term:

$$\mathcal{L}_{\text{balance}} = \alpha \sum_{i=1}^{N} \text{Load}(i)\, p_i,$$

where $N = 8$ is the number of experts and $\alpha = 0.4$. Here, $\text{Load}(i)$ denotes the proportion of tokens assigned to expert $i$ in the current batch, and $p_i$ represents the average gating probability allocated to expert $i$. Minimizing this loss encourages the distribution of expert loads to approach a uniform distribution, thus preventing a few experts from being overutilized while others remain idle. The hyperparameter $\alpha$ is used to balance the importance of the load-balancing objective against the primary task loss.

The model employs a vocabulary size of 50,257 (for GPT-2 tokenizer compatibility), a maximum sequence length of 2,048, three Transformer layers, and a final linear projection that maps hidden states to vocabulary logits. Training is conducted on the Penn Treebank (PTB) dataset (Marcinkiewicz, 1994), where the corpus is tokenized using the GPT-2 tokenizer and sequences are padded or truncated to 2,048 tokens. Optimization is performed with AdamW at a learning rate of $5 \times 10^{-5}$, using a linear warm-up of 1,000 steps. To stabilize training, gradient clipping with a maximum norm of 1.0 and gradient accumulation over four steps are employed. The effective batch size is four sequences per step, and models are trained for up to 100 epochs with early stopping based on validation loss, using a patience of three epochs.

During training, we jointly monitor the cross-entropy loss and the load-balancing term, while also analyzing expert routing behavior. In particular, we record the routing distributions across experts, the per-layer utilization statistics, and the temporal dynamics of load balancing. These analyses provide insights into the specialization stability of experts and the overall effectiveness of the proposed routing mechanism.

### A.1.2 EXPERIMENT ON THE PRE-TRAINING PHASE

We conduct pretraining experiments across multiple Mixture-of-Experts (MoE) architectures to systematically evaluate the effect of different model configurations on language modeling performance. Specifically, we examine three backbone structures, Mixtral-MoE (Jiang et al., 2024), Llama-MoE (Zhu et al., 2024), and GPT-2-MoE (Lagler et al., 2013), each augmented with varying numbers of experts and layers.

For the Mixtral-MoE and the Llama-MoE configuration, we train models with three MoE layers and either 4 or 8 experts per layer. These models contain 150M and 200M total parameters, respectively, with approximately 135M active parameters during inference. The detailed architectural and training settings follow those outlined in Appendix A.1.1.

Our GPT2-MoE model builds upon the GPT-2 architecture by incorporating Mixture-of-Experts (MoE) layers into the Transformer blocks. Each block consists of LayerNorm, causal self-attention, and an MoE feed-forward subnetwork, where the conventional MLP is replaced with eight experts, each implemented as a two-layer feed-forward network with GELU activation. For every token, the router selects the top-2 experts and assigns normalized weights through a softmax function. To ensure balanced utilization of experts, a load-balancing loss is added to the objective with a coefficient of 0.4. The overall model follows an autoregressive design with twelve Transformer layers, twelve attention heads, a hidden dimension of 768, and a maximum sequence length of 1024. Token and positional embeddings are used at the input, while the output head is weight-tied with the token embedding.

The training data is stored in memory-mapped binary format containing tokenized sequences for both training and validation. Each batch is constructed by randomly slicing fixed-length sequences, eliminating the need for padding or attention masks. The training objective combines the standard language modeling cross-entropy loss with the auxiliary load-balancing loss, while validation is performed solely on the language modeling component to provide a clean measure of generalization. Optimization is carried out using AdamW, with weight decay applied to weight parameters but excluded for biases and LayerNorm parameters. The learning rate starts from $6 \times 10^{-4}$, warms up for the first 2000 steps, and then decays following a cosine schedule to a minimum of $6 \times 10^{-5}$. Gradient clipping at 1.0 is applied to stabilize training. To achieve larger effective batch sizes, we adopt gradient accumulation over 40 steps by default, which scales automatically when distributed training is enabled. Training supports both float16 and bfloat16 mixed precision with automatic gradient scaling.

All Mixtral-MoE and Llama-MoE models are pretrained on the Penn Treebank (PTB) (Marcinkiewicz, 1994) and WikiText-2 (Merity et al., 2016) datasets. For the GPT-2-MoE models, we pretrain on OpenWebText (Gokaslan et al., 2019), a large-scale corpus designed to approximate the distribution of the original GPT-2 training data, in order to assess scalability under web-scale data. All pretraining experiments are conducted on the task of language modeling, where the objective is to minimize the token-level cross-entropy loss with an additional load-balancing regularization term for MoE routing.

Table 5: Ablation study on memory-related hyperparameters. Moderate values of the memory impact factor ($\alpha = 0.5$) and buffer size (128) yield the lowest PPL. Extremes at either end of these hyperparameters result in degraded performance, highlighting the necessity of a balanced configuration for optimal routing effectiveness.

| Memory-related hyperparameters | Memory Impact Factor $\alpha$ | | | | Buffer Size | | |
| --- | --- | --- | --- | --- | --- | --- | --- |
| | 0 | 0.2 | 0.5 | 0.8 | 64 | 128 | 256 |
| PPL | 74.48 | 73.82 | 69.68 | 74.02 | 75.64 | 69.68 | 71.32 |

The choice of datasets is aligned with the scale of the models under consideration. PTB and WikiText-2 are relatively small dataset, which makes them suitable for medium-sized models, where training efficiency and rapid experimentation are prioritized. In contrast, larger GPT-2-MoE models require exposure to significantly more diverse linguistic patterns to realize their capacity. Therefore, we employ OpenWebText, a 40 GB corpus with web-scale coverage, to ensure that these models can fully exploit their parameterization and demonstrate scalability.

In the comparative experiments, both the baseline models and those enhanced with Memory-Aware Routing (MAR) were trained under an identical load-balancing strategy, as specified in Appendix A.1.1. This ensures that any observed differences in performance can be attributed solely to the introduction of MAR rather than variations in routing regularization. For all MAR-augmented models, the buffer size was consistently fixed at 128, and the memory impact factor was set to 0.5, providing a stable configuration for evaluating the effectiveness of memory-guided routing.

### A.1.3  EXPERIMENT ON THE FINE-TUNING PHASE

For fine-tuning, we initialize with the open-source OLMoE 1B-7B model (Muennighoff et al., 2024) and conduct experiments with and without MAR across four representative downstream tasks. The datasets used for each task are summarized below:

- **Language Modeling:** Penn Treebank (PTB) (Marcinkiewicz, 1994). PTB contains approximately one million words from the Wall Street Journal and is a widely used benchmark for evaluating language modeling performance. We use perplexity (PPL) as the evaluation metric.
- **Knowledge Application:** Massive Multitask Language Understanding (MMLU) (Hendrycks et al., 2021). MMLU covers multiple domains including humanities, science, and professional knowledge, testing the model's ability to recall and apply factual information.
- **Commonsense Reasoning:** BIG-bench Hard (BBH) (Suzgun et al., 2022). BBH consists of challenging problems that require multi-step reasoning and knowledge beyond memorization, evaluating the model's common-sense reasoning ability.
- **Mathematical Reasoning:** GSM8K (Cobbe et al., 2021) and SVAMP (Patel et al., 2021). GSM8K is a collection of grade-school-level math word problems, while SVAMP evaluates the robustness and generalization of arithmetic reasoning.

Each data set is pre-processed to be compatible with the OLMoE tokenizer, and standard train-validation splits are used. This setup allows us to systematically assess the impact of MAR on both general language modeling and task-specific reasoning performance.

### A.1.4  ABLATION STUDY

To further analyze the key design choices of Memory Aware Routing, we conduct ablation experiments from two perspectives: the influence parameter $\alpha$ of the interest distribution and the memory buffer size. The results are shown in Table 5.

The ablation study is conducted on the Mixtral-MoE model with three MoE layers, each containing four experts. The experiments are performed on the PTB dataset (Marcinkiewicz, 1994), focusing on

language modeling. For consistency, the load balancing strategy and the load balancing weight are set as in Appendix A.1.1. All other model hyperparameters, such as hidden size, number of attention heads, and maximum sequence length, are kept identical to the base configuration. Training is performed on the same hardware setup as described in Appendix A.1.1, ensuring that comparisons reflect only the effect of the ablated components. This study aims to isolate the contribution of individual mechanisms within the Mixtral-MoE architecture and to quantify their impact on model performance.

We first examine the effect of $\alpha$ on model performance. When $\alpha = 0$, the model relies solely on instantaneous features, leading to severe pseudo-balance and insufficient expert specialization. As $\alpha$ increases from 0.2 to 0.5, the perplexity (PPL) decreases by 5%, indicating that incorporating moderate historical interest effectively breaks pseudo-balance and accelerates the convergence of expert specialization. The best performance is achieved at $\alpha = 0.5$, where instantaneous features and long-term preferences reach an optimal balance. However, further increasing $\alpha$ to 0.8 degrades performance by 6%, suggesting that excessive reliance on historical interests introduces expert "path dependence" and overfitting to past patterns, thereby weakening generalization ability.

Next, we investigate the impact of buffer size on modeling the interest distribution. With a small buffer (64), too few historical samples are retained, making the interest distribution sensitive to short-term fluctuations and leading to unstable expert specialization. At a buffer size of 128, historical and instantaneous information are better balanced: short-term noise is suppressed without introducing stale information, resulting in the best performance. When the buffer size increases to 256, although more history is preserved, the interest distribution is diluted by outdated samples, reducing the model's sensitivity to current inputs and ultimately degrading performance by approximately 2%.

In summary, the combination of moderate historical memory and instantaneous features is crucial for the effectiveness of MAR. In our experiments, the optimal configuration is achieved at $\alpha = 0.5$ and a buffer size of 128. These findings validate the soundness of our design and further highlight that a proper balance between short-term features and long-term preferences is key to efficient expert specialization and stable model convergence.

## A.2 MORE ANALYSIS

### A.2.1 THE PSEUDO-BALANCE PHENOMENON

As depicted in Figure 6, the load-balanced model demonstrates significantly less sensitivity to expert masking. Its perplexity remains low even when a considerable portion of its top experts are disabled, which indicates a high degree of redundancy. Conversely, the baseline model, which lacks load balancing, experiences a much steeper decline in performance. This confirms the successful specialization and non-interchangeability of its experts. Thus, these results provide direct evidence that expert balancing methods lead to substantial knowledge overlap.

To investigate the pseudo-balancing phenomenon, we compared the training dynamics of models with and without load balancing. Figure 7 indicates that without the load-balancing regularization, the routing network tends to activate only a small subset of experts, leaving the majority largely unused and resulting in severe load imbalance. With the introduction of the load-balancing regularizer, Figure 8 shows that expert activations become more uniform, achieving an apparent global balance. However, this superficial balance impedes the formation of stable expert specialization, highlighting the inherent tension between expert-centered load balancing and expert differentiation—a fundamental challenge in MoE training.

To quantify this effect, we measured the standard deviation of expert utilization rates, capturing the uniformity of each expert's activation across all training tokens. Table 6 shows that without load balancing, the expert utilization standard deviation is approximately 0.8, whereas adding the regularizer reduces it to around 0.2, demonstrating a significant improvement in expert usage uniformity.

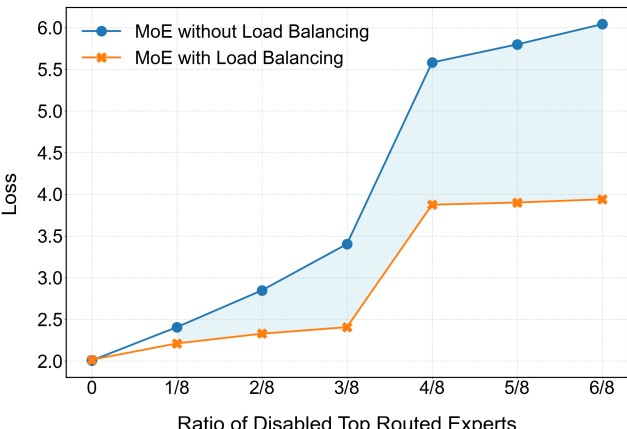

Figure 6: Loss across varying ratios of disabled top-routed experts. Models without load balancing show greater sensitivity, suggesting that load balancing amplifies redundancy among routed experts.

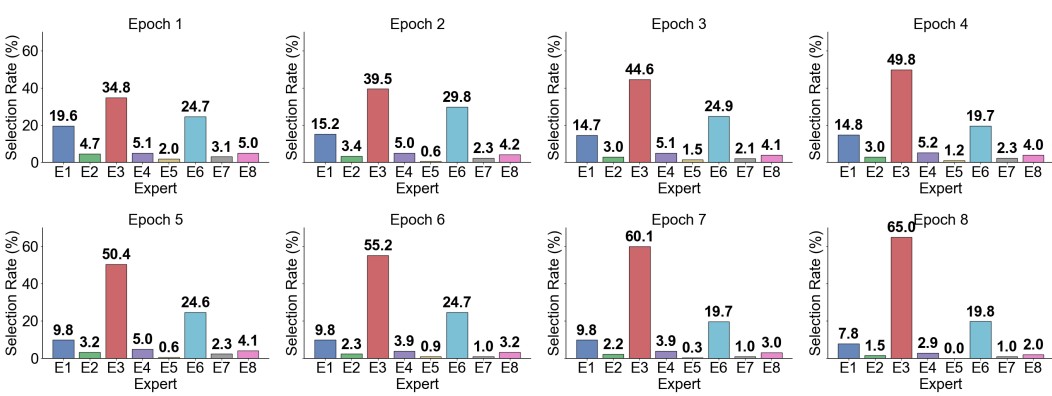

Figure 7: Expert load distribution across training epochs in the model without LBL. Without LBL, the model tends to activate only a small subset of experts, and this imbalance becomes progressively more severe over the course of training.

### A.2.2 EXPERT SELECTION OF MAR

Figure 9 illustrates that while standard load-balanced models show input assignments constantly oscillating between experts even late in training, models with MAR maintain stable expert assignments. This directly shows that MAR effectively solves the pseudo-balance problem.

### A.2.3 EFFECTIVENESS OF MAR IN MAINTAINING BALANCE

We further analyzed the effectiveness of Memory-Aware Routing (MAR) in maintaining expert load balance on two additional base models: Llama-MoE and GPT2-MoE. The results, shown in Figure 10 and Figure 11, indicate that MAR also achieves stable and consistent expert utilization with a low standard deviation on these models.

For the evaluation metric, we also adopt the standard deviation of expert utilization rates to measure the degree of load balance across experts. As shown in Table 7, models with MAR maintain an expert utilization standard deviation of approximately 2.57. Although this value is slightly higher than that achieved by conventional load-balancing strategies, we argue that it more faithfully reflects the true distribution of the data. Forcing expert utilization to be perfectly uniform would result in some tokens being assigned to experts with inconsistent semantics, thereby introducing pseudo-balance. In contrast, the moderate deviation observed under MAR indicates that experts are specializing

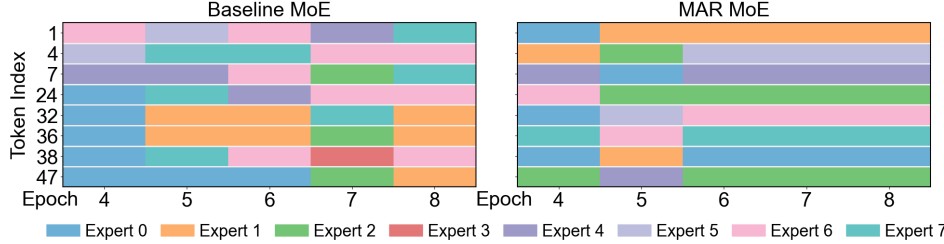

Figure 8: Expert load distribution across training epochs in the model with LBL. Under the constraint of LBL, the model successfully balances the utilization of experts, leading to a more even load distribution.

Figure 9: Expert routing stability comparison between baseline Mixtral-MoE and Mixtral-MoE+MAR. This figure compares how tokens are routed to experts in the training epochs. The Baseline MoE shows unstable routing, as the same tokens are sent to different experts. In contrast, MAR MoE has a stable pattern, showing successful expert specialization.

according to the semantic characteristics of the data, which is essential for achieving meaningful specialization and avoiding artificial pseudo-balance.

### A.2.4 EVALUATION OF MAR OVERHEAD

To substantiate our claim that MAR introduces negligible overhead, we provide a quantitative comparison of GPU memory usage and routing latency between LBL and LBL+MAR, as summarized in Table 8. The results indicate that MAR incurs only a minor increase in peak GPU memory (+0.3 GB) and routing latency (+0.8 ms), confirming that its computational overhead is minimal.

### A.2.5 MEMORY BUFFER UPDATE STRATEGIES

We conducted additional experiments comparing FIFO, LRU, LFU, and RAND. Table 9 indicates that FIFO achieves nearly identical PPL to the strongest baselines and consistently ranks among the top two across datasets, while being substantially more efficient—approximately 2–3× faster than LRU/LFU. These findings suggest that FIFO provides the most favorable quality–efficiency trade-off in large-scale training.

Regarding the similarity metric, we adopt cosine similarity due to its scale-invariant property, which ensures more stable behavior in high-dimensional embeddings. Additionally, it incurs lower computational cost compared to Euclidean distance, aligning well with our efficiency-oriented setting. EMA-based, attention-weighted, or representational update schemes introduce additional computations—such as similarity evaluation, ranking, or sampling—at every forward pass. At our training scale, such complexity would substantially reduce throughput and is therefore impractical. Conse-

**Expert Selection Rates per Epoch (Llama-MoE+MAR)**

Figure 10: Expert load distribution across training epochs in the Llama-MoE+MAR model. Models with MAR exhibit stable and balanced utilization of experts, indicating the effectiveness of Memory-Aware Routing in mitigating expert collapse.

**Expert Selection Rates per Epoch (GPT2-MoE+MAR)**

Figure 11: Expert load distribution across training epochs in the GPT2-MoE+MAR model. Models with MAR exhibit stable and balanced utilization of experts, indicating the effectiveness of Memory-Aware Routing in mitigating expert collapse.

quently, we employ the lightweight and effective FIFO strategy, as validated by our experiments.

### A.2.6 EFFICIENT IMPLEMENTATION OF MEMORY-AWARE ROUTING

In the main text, updating each expert's preference vector by aggregating features from its buffer has complexity $\mathcal{O}(Nd)$ per expert, where $N$ is the buffer size and $d$ is the hidden dimension. However, this computation can be optimized to $\mathcal{O}(1)$ per update using a running sum. Specifically, for expert $i$ with buffer $\mathcal{B}_i$, let the current preference vector be

$$d_i = \frac{1}{|\mathcal{B}_i|} \sum_{h \in \mathcal{B}_i} h.$$

When a new token representation $h_{\text{new}}$ is appended and the oldest entry $h_{\text{old}}$ is evicted, the preference vector can be updated incrementally as

$$d_i \leftarrow d_i + \frac{1}{|\mathcal{B}_i|}(h_{\text{new}} - h_{\text{old}}),$$

Table 6: Expert load standard deviation of Mixtral-MoE with LBL and without LBL.

| Method | Epoch 1 | Epoch 2 | Epoch 3 | Epoch 4 | Epoch 5 | Epoch 6 | Epoch 7 | Epoch 8 | Avg |
|--------|---------|---------|---------|---------|---------|---------|---------|---------|-----|
| Load Balance | 0.0857 | 0.0696 | 0.0331 | 0.0484 | 0.0829 | 0.0500 | 0.0866 | 0.0500 | 0.0633 |
| No Balance | 12.00 | 15.65 | 16.35 | 19.71 | 19.16 | 22.04 | 24.63 | 27.60 | 19.14 |

Table 7: Expert load standard deviation of Mixtral-MoE without LBL, with LBL and with MAR. LBL is the standard load balancing loss, and MAR is our proposed Memory-Aware Routing. The results show that MAR significantly decreases the load standard deviation, indicating more equal expert utilization.

| Method | Epoch 1 | Epoch 2 | Epoch 3 | Epoch 4 | Epoch 5 | Epoch 6 | Epoch 7 | Epoch 8 | Avg |
|--------|---------|---------|---------|---------|---------|---------|---------|---------|-----|
| w/o LBL | 12.00 | 15.65 | 16.35 | 19.71 | 19.16 | 22.04 | 24.63 | 27.60 | 19.14 |
| LBL | 0.0857 | 0.0696 | 0.0331 | 0.0484 | 0.0829 | 0.0500 | 0.0866 | 0.0500 | 0.0633 |
| LBL+MAR | 2.63 | 2.39 | 2.62 | 2.71 | 2.52 | 2.58 | 2.45 | 2.63 | 2.57 |

avoiding a full summation over the buffer. This ensures that each update has constant time complexity $\mathcal{O}(d)$, independent of the buffer size $N$, enabling scalable training even with very large memory buffers. Similarly, token–expert similarity computations remain $\mathcal{O}(Kd)$ per token, and no additional memory beyond the buffer itself is required. This efficient implementation allows MAR to be applied in large-scale experiments with long buffers without introducing significant training overhead.

Table 8: Comparison of LBL and LBL+MAR in terms of GPU Memory Usage and Routing Latency. Measured on NVIDIA RTX 4090, with batch size = 16. Peak GPU memory and average routing latency (average time per forward pass, in ms) are taken at the same number of training steps.

| Metric | LBL | LBL + MAR | Absolute change |
|---|---|---|---|
| Peak GPU memory (GB) | 12.3 | 12.6 | +0.3 |
| Training Routing latency (ms) | 7.8 | 8.6 | +0.8 |

Table 9: Comparison of memory buffer update strategies. FIFO ranks among the top two strategies on both datasets while achieving substantially higher training efficiency (approximately 2–3× faster than LRU/LFU). Time is measured under the same number of training steps.

| Dataset | Strategy | Loss | PPL | Time (min) |
|---|---|---|---|---|
| PTB | **FIFO** | **3.7374** | **41.99** | **137.0** |
| | LRU | 3.7371 | 41.98 | 313.8 |
| | LFU | 3.7601 | 42.95 | 302.0 |
| | RAND | 3.7507 | 42.55 | 150.4 |
| WikiText-2 | **FIFO** | **3.8044** | **44.90** | **79.4** |
| | LRU | 3.8123 | 45.26 | 169.1 |
| | LFU | 3.8011 | 44.75 | 165.3 |
| | RAND | 3.8157 | 45.41 | 81.6 |