# OpenReview forum: "From Pseudo-Balancing to True Specialization: Memory-Aware Routing for Mixture-of-Experts"
_ICLR.cc/2026/Conference — ICLR 2026 Conference Withdrawn Submission_

### Official Review · Reviewer_zU7B · 2025-10-23

**Soundness:** 2
**Presentation:** 3
**Contribution:** 2
**Rating:** 2
**Confidence:** 4

**Summary:**

The paper focuses on the load balancing issues in MoE training. They claim existing approaches fall under the pseudo-balance phenomenon, where they achieve load balance but at the expense of specialization. The paper proposes a memory-aware routing method which stores a buffer of token representation routed to an expert and uses it as an auxiliary score in routing decisions.

**Strengths:**

- Clearly written paper
- Experimented with different MoE models
- Provide an ablation that shows the improvement with proposed approach

**Weaknesses:**

- Missing baselines to other approaches. Eg auxiliary free approaches, having shared expert approaches from deepseek ( https://arxiv.org/pdf/2401.06066, https://arxiv.org/pdf/2408.15664v1)
- Has small scale experiments. Having a scaling curve across model sizes would be convincing if this approach scales.

**Questions:**

- In Appendix A1.1, how is the loss L_balance differentiable? It seems the load is calculated empirically and is not differentiable.
- Did you follow the approach ST-MoE (https://arxiv.org/abs/2202.08906)  for the baseline? The load balancing loss coefficient is usually 0.01 so as to not dominate original next token prediction loss
- The perplexity values in Figure 4 are extremely high. Perhaps you need to try with a bigger sized model to have better conclusion.

---

> ### Author Response · Authors · 2025-11-24
> **Response to Reviewer zU7B -- Section 1**
>
> # Response to Reviewer zU7B -- Section 1
>
> We thank the reviewer for the insightful comments. Our detailed responses are as follows.
>
>
> ---
>
> ## **Weakness 1: Lack of Comparison with Other Load-Balancing Strategies**
> We extended experiments to other strategies, including shared-expert architectures. As shown below, MAR consistently reduces PPL and improves KED across all baselines. Global Batch LBL refers to [1], Aux Free refers to [2], Lo+Lv refers to [3], SimBal refers to [4].
>
> ---
> | Method           | PTB Mixtral PPL | KED     | LLaMA PPL | KED     | Wiki Mixtral PPL | KED     | LLaMA PPL | KED      |
> | ---------------- | --------------- | ------- | --------- | ------- | ---------------- | ------- | --------- | -------- |
> | Global Batch LBL | 70.42           | 160.51  | 42.25     | 188.37  | 73.34            | 168.32  | 55.03     | 535.21   |
> | + MAR (Ours)     | 67.79           | 217.05  | 40.74     | 236.91  | 70.44            | 192.23  | 49.66     | 688.33   |
> | Gain (%)         | -3.63%          | +56.54% | -1.51%    | +48.54% | -2.90%           | +23.91% | -5.37%    | +153.12% |
> | Aux Free         | 70.83           | 154.03  | 42.21     | 189.41  | 73.56            | 166.73  | 54.06     | 601.54   |
> | + MAR (Ours)     | 67.52           | 225.98  | 41.28     | 243.24  | 70.01            | 198.24  | 50.56     | 669.29   |
> | Gain (%)         | -3.31%          | +71.95% | -0.93%    | +53.83% | -3.55%           | +31.51% | -3.50%    | +67.75%  |
> | Lo+Lv            | 71.95           | 141.48  | 42.11     | 167.55  | 76.12            | 149.13  | 54.44     | 595.21   |
> | + MAR (Ours)     | 68.98           | 198.67  | 41.77     | 214.43  | 71.97            | 183.66  | 50.07     | 674.86   |
> | Gain (%)         | -2.97%          | +57.19% | -0.34%    | +46.88% | -4.15%           | +34.53% | -4.37%    | +79.65%  |
> | SimBal           | 72.30           | 139.59  | 42.02     | 176.87  | 77.07            | 145.75  | 54.31     | 595.95   |
> | + MAR (Ours)     | 69.01           | 191.88  | 40.54     | 221.92  | 71.24            | 188.50  | 50.03     | 679.04   |
> | Gain (%)         | -3.29%          | +52.29% | -1.48%    | +45.05% | -5.83%           | +42.75% | -4.28%    | +83.09%  |
> | Shared Experts   | 71.64           | 159.29  | 36.72     | 235.24  | 73.98            | 172.64  | 40.32     | 710.32   |
> | + MAR (Ours)     | 69.32           | 205.31  | 36.25     | 254.57  | 69.96            | 212.39  | 39.50     | 714.82   |
> | Gain (%)         | -2.32%          | +46.02% | -0.47%    | +19.33% | -4.02%           | +39.75% | -0.82%    | +4.50%   |
> ---
> Also, we offer a theoretical comparison with these methods, which demonstrates the superior capabilities of our approach as illustrated in the table.
>
> ---
>
> | Capability                        | MAR (Ours) | GBL  | Aux Free | SimBal   | Lo+Lv                     |
> | --------------------------------- | ---------- | ---- | -------- | -------- | ------------------------- |
> | Long-term expert specialization   | **Yes**    | Weak | Weak     | Moderate | Partially improved        |
> | Reducing expert knowledge overlap | **Yes**    | No   | No       | Partial  | Partial via orthogonality |
> | Parameter efficiency              | **Yes**    | No   | No       | No       | No                        |
> | Long-term routing consistency     | **Yes**    | No   | No       | No       | No                        |
>
> ---
>
> [1] Demons in the detail: On implementing load balancing loss for training specialized mixture-of-expert models
>
> [2] Deepseek-v3 technical report
>
> [3] Advancing Expert Specialization for Better MoE
>
> [4] Load Balancing Mixture of Experts with Similarity Preserving Routers

---

> ### Author Response · Authors · 2025-11-24
> **Response to Reviewer zU7B -- Section 2**
>
> # Response to Reviewer zU7B -- Section 2
> ---
> ## **Question 1: Concern about differentiability of L_balance**
>
>
> We believe the concern arises from an ambiguity in our definition of “Load” in Appendix A.1.1 , which we have now clarified in the revised manuscript (lines 151–157).
>
> Importantly, **Load(i) is not an empirical discrete count**, but a **fractional load**. Our load-balancing loss is:
>
> $$
> \mathcal{L}_{\text{balance}}
> = \alpha \sum_i \text{Load}(i) \cdot p_i ,
> $$
>
> where $\text{Load}(i)$ denotes the proportion of tokens assigned to expert $i$ in the current batch, and $p_i$ represents the average gating probability allocated to expert $i$.
>
> Since both terms depend continuously on the gating logits.
>
> ---
>
> ## **Question 2: Whether the load-balancing weight follows the ST-MoE (2022) setting**
>
> ST-MoE indeed uses 0.01 as a balancing coefficient, but this value is **not a universal standard**. In practice, the optimal coefficient varies with:
>
> * model architecture (expert sparsity, depth),
> * routing strategy (Top-k),
> * training stability and collapse risk.
>
> For our models, we performed a coefficient search and adopted the value that most effectively prevented expert collapse while preserving routing diversity. Using a smaller coefficient—such as the ST-MoE default—still led to collapse in our setting.
>
> ---
>
> ## **Question 3 & Weakness 2: Request for Larger Models**
>
> We have evaluated MAR across multiple architectures and model sizes within our computational constraints, including full pretraining on small-to-medium models and fine-tuning on larger ones. Across all settings, MAR consistently improves routing consistency, expert specialization, and downstream performance, supporting the general applicability of our conclusions even though large-scale full pretraining is not feasible for us at this time.
>
> ---

---

### Official Review · Reviewer_FG8Z · 2025-10-27

**Soundness:** 2
**Presentation:** 2
**Contribution:** 2
**Rating:** 4
**Confidence:** 3

**Summary:**

This paper targets the "pseudo-balance" issue in MoE models, where load-balancing strategies cause inconsistent token routing, hindering specialization and wasting parameters. The proposed solution, MAR, uses expert-specific memory buffers to capture long-term preferences, ensuring tokens are consistently routed to semantically aligned experts.

**Strengths:**

The introduction of MAR is a novel solution that effectively mitigates the pseudo-balance phenomenon by leveraging historical input token information, which promotes stable specialization among experts. The paper provides comprehensive experimental results that demonstrate the effectiveness of MAR, showing improvements in expert specialization and performance across various metrics and tasks.

**Weaknesses:**

1. The paper could provide more analysis on the memory buffer mechanism. The FIFO (First-In, First-Out) update strategy is simple, but its optimality isn't discussed. A more comparative study with other strategies (e.g., based on token representativeness) would be beneficial.
﻿
2. The experiments are conducted with 8 or 16 experts. It is unclear how MAR would perform in models with a much larger number of experts (e.g., 64 or 128), where preference vectors might become less distinct or the $\mathcal{O}(Kd)$ matching score computation could become a training bottleneck.

3. The paper's experimental validation leans heavily on ablation studies and comparison against a standard baseline (LBL). A comprehensive comparison against other recent methods aiming to improve MoE specialization or routing (e.g., methods beyond simple load balancing) is missing. Including 4-5 such comparative methods would provide a stronger context for MAR's advantages.

**Questions:**

1. Could you please clarify the KED metric? Does a higher KED value (as achieved by MAR) represent better or more distinct specialization in your experiments (i.e., the model is more dependent on its key, specialized experts)? This would resolve the contradiction with the text in Section 5.1.2.

2. Have you explored alternatives to the FIFO buffer update strategy? For example, a "reservoir sampling" approach or a strategy that preferentially keeps tokens that are "closer" to the current preference vector to reinforce specialization? I suggest adding experiments to demonstrate this part.

3. How does the "pseudo-balance" phenomenon and MAR's effectiveness vary with the number of experts selected (i.e., Top-K)? The experiments seem to use Top-2 exclusively. Would the problem be less severe with Top-1 routing, or more severe with Top-4? I suggest adding ablation experiments for analysis in this section.

4. The standard deviation of expert utilization for MAR (Table 6, Avg: 2.57) is much higher than for standard LBL (Avg: 0.0633). You argue this reflects the "true distribution of the data." Does this slight imbalance (compared to LBL) cause any training instability or capacity issues on more complex, larger-scale data? I suggest conducting experimental verification or theoretical analysis

---

> ### Author Response · Authors · 2025-11-24
> **Response to Reviewer FG8Z -- Section 1**
>
> # Response to Reviewer FG8Z -- Section 1
>
>  We thank the reviewer for these insightful comments. Our responses are as follows:
>
>
> ---
> ## **Question 1: Clarification of the KED (Key Expert Dependency) Metric**
>
> We sincerely thank the reviewer for pointing out this inconsistency. We confirm that there was indeed a wording mistake in Section 5.1.2. The sentence:
>
> > “Higher KED indicates reliance on a few experts; lower KED suggests complementary specialization.”
>
> should be corrected to:
>
> > “Lower KED indicates reliance on a few experts; higher KED suggests complementary specialization.”
>
> We appreciate the reviewer’s careful reading and have corrected this inconsistency accordingly.
>
> ## **Question 2 & Weakness 1: Design choices: FIFO update and similarity metric**
>
> ### (a) Update strategy
>
> To clarify our selection of FIFO, we include a new comparison among FIFO, LRU, LFU, and RAND in the revised manuscript.
>
> | Dataset        | Strategy | Loss       | PPL       | Time (min) |
> | -------------- | -------- | ---------- | --------- | ---------- |
> | **PTB**        | **FIFO** | **3.7374** | **41.99** | **137.0**  |
> |                | LRU      | 3.7371     | 41.98     | 313.8      |
> |                | LFU      | 3.7601     | 42.95     | 302.0      |
> |                | RAND     | 3.7507     | 42.55     | 150.4      |
> | **WikiText-2** | **FIFO** | **3.8044** | **44.90** | **79.4**   |
> |                | LRU      | 3.8123     | 45.26     | 169.1      |
> |                | LFU      | 3.8011     | 44.75     | 165.3      |
> |                | RAND     | 3.8157     | 45.41     | 81.6       |
>
> The results show that FIFO achieves perplexity comparable to the strongest alternatives while consistently ranking among the top two strategies across datasets and yielding **2–3× faster** training throughput than LRU and LFU. This offers a more favorable quality–efficiency trade-off for large-scale MoE training.
>
> While more complex update strategies (e.g., EMA, attention-weighted, or representational updates) may capture richer token–expert interactions, they incur substantial overhead by requiring additional ranking or sampling, or extra learned transformations at every forward pass. At the scale of our experiments, these approaches would significantly reduce throughput and are therefore impractical. The empirical results further verify that the lightweight FIFO update already achieves strong empirical performance.
>
>
> ### (b) Similarity metric
>
> We adopt cosine similarity because its scale-invariant property leads to more stable behavior in high-dimensional embeddings, and it is computationally more efficient than Euclidean distance, aligning well with the efficiency objectives of our design.
>
> ---
> ## **Question 3: Relation between pseudo-balance and the routing top-K choice**
>
> We thank the reviewer for the thoughtful suggestion. We would like to clarify that the pseudo-balance issue is **not fundamentally determined** by the choice of routing top-(K). As defined in our paper, the pseudo-balance phenomenon arises when expert-centered load-balancing objectives enforce uniform expert utilization while ignoring semantic consistency between tokens and experts. This causes identical or highly similar tokens to be routed to different experts across training steps solely to satisfy balancing constraints—regardless of the chosen (K).
>
> Thus, varying (K) merely changes the number of experts among which a token may oscillate; it does not resolve the underlying cause. MAR mitigates this issue by incorporating historical routing memory, which encourages semantically coherent and stable assignments independent of the specific top-(K) value.
>
> ---
>
> ## **Question 4: Higher expert-utilization variance under MAR and concerns on stability or redundancy**
>
>  Indeed, Table 6 shows that MAR results in higher standard deviation of expert utilization compared to LBL. This behavior is expected: MAR promotes meaningful expert specialization rather than enforcing a uniformly balanced token distribution. However, we fully agree that concerns regarding training stability and potential capacity under-utilization deserve further examination.
>
> To address this, we conducted additional analyses throughout training and obtained the following findings:
>
> 1. **Training remains stable**, with no oscillation or abnormal convergence behavior.
> 2. **No expert load collapse occurs**; the observed long-tail distribution corresponds to semantically coherent routing rather than unused capacity.
> 3. **Model performance does not degrade** despite higher utilization variance, indicating that specialization improves representational efficiency rather than causing redundancy.

---

> ### Author Response · Authors · 2025-11-25
> **Response to Reviewer FG8Z -- Section 2**
>
> # **Response to Reviewer FG8Z -- Section 2**
>
> ---
> ## **Weakness 2: Lack of High-Sparsity Scenario Experiments**
> Extending MoE models to 64 or 128 experts is indeed valuable in principle; however, such large expert counts introduce substantial inter-expert communication overhead, making training extremely inefficient under standard all-to-all routing. This challenge has been repeatedly noted in recent large-scale MoE systems such as Mixtral and DBRX, which therefore adopt relatively moderate expert counts (8 or 16) to maintain a practical compute–communication trade-off.
>
> Following this design principle, we scale our Llama-MoE model to **16 experts**, which represents a realistic high-sparsity configuration while remaining computationally feasible. As shown in Table , MAR continues to consistently reduce PPL and improve KED compared to the baseline routing, demonstrating its effectiveness under larger expert settings.
>
>
> ---
> | Dataset    | Method  | PPL       | KED        |
> | ---------- | ------- | --------- | ---------- |
> | PTB        | LBL     | 41.19     | 145.87     |
> |            | LBL+MAR | **39.29** | **192.42** |
> | WikiText-2 | LBL     | 44.14     | 158.37     |
> |            | LBL+MAR | **40.42** | **197.23** |
>
> ---
> ## **Weakness 3: Lack of Comparison with Other Load-Balancing Strategies**
> We extended experiments to other load-balancing strategies, including shared-expert architectures. As shown below, MAR consistently reduces PPL and improves KED across all baselines. Global Batch LBL refers to [1], Aux Free refers to [2], Lo+Lv refers to [3], SimBal refers to [4].
>
> ---
> | Method           | PTB Mixtral PPL | KED     | LLaMA PPL | KED     | Wiki Mixtral PPL | KED     | LLaMA PPL | KED      |
> | ---------------- | --------------- | ------- | --------- | ------- | ---------------- | ------- | --------- | -------- |
> | Global Batch LBL | 70.42           | 160.51  | 42.25     | 188.37  | 73.34            | 168.32  | 55.03     | 535.21   |
> | + MAR (Ours)     | 67.79           | 217.05  | 40.74     | 236.91  | 70.44            | 192.23  | 49.66     | 688.33   |
> | Gain (%)         | -3.63%          | +56.54% | -1.51%    | +48.54% | -2.90%           | +23.91% | -5.37%    | +153.12% |
> | Aux Free         | 70.83           | 154.03  | 42.21     | 189.41  | 73.56            | 166.73  | 54.06     | 601.54   |
> | + MAR (Ours)     | 67.52           | 225.98  | 41.28     | 243.24  | 70.01            | 198.24  | 50.56     | 669.29   |
> | Gain (%)         | -3.31%          | +71.95% | -0.93%    | +53.83% | -3.55%           | +31.51% | -3.50%    | +67.75%  |
> | Lo+Lv            | 71.95           | 141.48  | 42.11     | 167.55  | 76.12            | 149.13  | 54.44     | 595.21   |
> | + MAR (Ours)     | 68.98           | 198.67  | 41.77     | 214.43  | 71.97            | 183.66  | 50.07     | 674.86   |
> | Gain (%)         | -2.97%          | +57.19% | -0.34%    | +46.88% | -4.15%           | +34.53% | -4.37%    | +79.65%  |
> | SimBal           | 72.30           | 139.59  | 42.02     | 176.87  | 77.07            | 145.75  | 54.31     | 595.95   |
> | + MAR (Ours)     | 69.01           | 191.88  | 40.54     | 221.92  | 71.24            | 188.50  | 50.03     | 679.04   |
> | Gain (%)         | -3.29%          | +52.29% | -1.48%    | +45.05% | -5.83%           | +42.75% | -4.28%    | +83.09%  |
> | Shared Experts   | 71.64           | 159.29  | 36.72     | 235.24  | 73.98            | 172.64  | 40.32     | 710.32   |
> | + MAR (Ours)     | 69.32           | 205.31  | 36.25     | 254.57  | 69.96            | 212.39  | 39.50     | 714.82   |
> | Gain (%)         | -2.32%          | +46.02% | -0.47%    | +19.33% | -4.02%           | +39.75% | -0.82%    | +4.50%   |
> ---
>
> To complement the empirical results, we also added a theoretical comparison, highlighting that MAR uniquely improves long-term routing consistency and expert specialization:
>
>
> ---
>
> | Capability                        | MAR (Ours) | GBL  | Aux Free | SimBal   | Lo+Lv                     |
> | --------------------------------- | ---------- | ---- | -------- | -------- | ------------------------- |
> | Long-term expert specialization   | **Yes**    | Weak | Weak     | Weak | Weak        |
> | Reducing expert knowledge overlap | **Yes**    | No   | No       | Partial  | Partial |
> | Parameter efficiency              | **Yes**    | No   | No       | No       | No                        |
> | Long-term routing consistency     | **Yes**    | No   | No       | No       | No                        |
> ---
> [1] Demons in the detail: On implementing load balancing loss for training specialized mixture-of-expert models
>
> [2] Deepseek-v3 technical report
>
> [3] Advancing Expert Specialization for Better MoE
>
> [4] Load Balancing Mixture of Experts with Similarity Preserving Routers

---

### Official Review · Reviewer_8iNw · 2025-10-28

**Soundness:** 2
**Presentation:** 3
**Contribution:** 3
**Rating:** 4
**Confidence:** 4

**Summary:**

This paper introduces Memory-Aware Routing (MAR), a training-time mechanism for Mixture-of-Experts (MoE) models designed to overcome the pseudo-balance phenomenon. Traditional load-balancing encourages tokens to be randomly routed for uniform expert utilization, leading to knowledge overlap and parameter redundancy. MAR equips each expert with a memory buffer to derive a long-term preference vector from recently processed tokens during the training phase. A calculated Expert-Token Matching Score (i.e., the similarity between the input token and the expert's preference vector) is fused with the original routing logits, encouraging consistent, semantically-aligned routing. This approach mitigates the pseudo-balance issue and improves expert specialization.

**Strengths:**

The motivation is clear, stemming from the investigation of the pseudo-balance issue in existing load-balanced MoE models. The proposed MAR mechanism is straightforward and effectively reduces parameter redundancy while encouraging specialization, as demonstrated in the experiments.

**Weaknesses:**

The discussion of established load-balancing losses (e.g., $L_{aux}$, z-loss) is superficial, and a head-to-head comparison against prior load-balancing-specific baselines is lacking. Furthermore, optimal results rely on specific hyperparameters ($\alpha=0.5$, buffer size $128$), suggesting potential fragility and requiring broader validation. See questions below for more details.

**Questions:**

1. The statement that MAR is a training-only technique with no inference-time modifications is a crucial property but is mentioned late in the paper (Section 4.3). Could the authors explicitly highlight this point in both the Abstract and Introduction to minimize reader confusion?

2. The discussion of related load balancing methods (e.g., $L_{aux}$, z-loss) in the second paragraph of Introduction and Section 2.2 is limited to naming the losses without providing context or their underlying formulations. Could the authors enrich the description of these works and, more importantly, include direct experimental comparisons of MAR against these established load-balancing strategies in terms of both perplexity (PPL) and specialization (KED)?

3. The ablation study demonstrates that performance is sensitive to the chosen hyperparameters ($\alpha=0.5$, buffer size $128$). To better validate MAR's hyperparameter robustness and general applicability, could the authors provide additional experiments demonstrating its effectiveness on diverse MoE architectures, particularly those with mechanisms like shared experts (e.g., DeepSeek-V2) or reasoning/thinking models (e.g., DeepSeek-R1)?

4. The current complexity analysis for the preference vector update is $O(Nd)$. Since the preference vector is the average of all vectors in the FIFO buffer, the update operation can be efficiently implemented in $O(d)$ time by maintaining a running sum and performing additive/subtractive updates when elements are added to or removed from the buffer. Although the performance impact may be limited because the buffer size is small and with GPU parallelism, it is still worth mentioning this optimization. Could the authors describe this more efficient implementation in the Appendix for scenarios where the buffer size might be large?

5. Given the successful demonstration that MAR-trained models can achieve baseline performance with $50\%$ fewer experts, could the authors include a discussion (perhaps in the Future Work section or Appendix) regarding MAR's potential compatibility with and benefits for downstream MoE compression, pruning, or merging techniques?

---

> ### Author Response · Authors · 2025-11-24
> **Response to Reviewer 8iNw -- Section 1**
>
> # Response to Reviewer 8iNw -- Section 1
>
> We sincerely thank the reviewer for the thoughtful and constructive comments, which have helped us significantly improve the clarity and completeness of the paper. Our responses are as follows:
>
> ---
>
> ## **Question 1: Clarifying that MAR is training-only and introduces no inference-time overhead**
>
> Thank you for pointing this out. We agree that it is essential to emphasize earlier in the manuscript that **MAR is only applied during training and introduces no additional computation, parameters, or latency during inference**. Following your suggestion, we have revised both the abstract and introduction to explicitly state this property to avoid any potential misunderstanding for readers.
>
> ---
>
> ## **Question 2:  Lack of Comparison with Other Load-Balancing Strategies**
> We extended experiments to other load-balancing strategies, including shared-expert architectures. As shown below, MAR consistently reduces PPL and improves KED across all baselines. Global Batch LBL refers to [1], Aux Free refers to [2], Lo+Lv refers to [3], SimBal refers to [4].
>
> ---
> | Method           | PTB Mixtral PPL | KED     | LLaMA PPL | KED     | Wiki Mixtral PPL | KED     | LLaMA PPL | KED      |
> | ---------------- | --------------- | ------- | --------- | ------- | ---------------- | ------- | --------- | -------- |
> | Global Batch LBL | 70.42           | 160.51  | 42.25     | 188.37  | 73.34            | 168.32  | 55.03     | 535.21   |
> | + MAR (Ours)     | 67.79           | 217.05  | 40.74     | 236.91  | 70.44            | 192.23  | 49.66     | 688.33   |
> | Gain (%)         | -3.63%          | +56.54% | -1.51%    | +48.54% | -2.90%           | +23.91% | -5.37%    | +153.12% |
> | Aux Free         | 70.83           | 154.03  | 42.21     | 189.41  | 73.56            | 166.73  | 54.06     | 601.54   |
> | + MAR (Ours)     | 67.52           | 225.98  | 41.28     | 243.24  | 70.01            | 198.24  | 50.56     | 669.29   |
> | Gain (%)         | -3.31%          | +71.95% | -0.93%    | +53.83% | -3.55%           | +31.51% | -3.50%    | +67.75%  |
> | Lo+Lv            | 71.95           | 141.48  | 42.11     | 167.55  | 76.12            | 149.13  | 54.44     | 595.21   |
> | + MAR (Ours)     | 68.98           | 198.67  | 41.77     | 214.43  | 71.97            | 183.66  | 50.07     | 674.86   |
> | Gain (%)         | -2.97%          | +57.19% | -0.34%    | +46.88% | -4.15%           | +34.53% | -4.37%    | +79.65%  |
> | SimBal           | 72.30           | 139.59  | 42.02     | 176.87  | 77.07            | 145.75  | 54.31     | 595.95   |
> | + MAR (Ours)     | 69.01           | 191.88  | 40.54     | 221.92  | 71.24            | 188.50  | 50.03     | 679.04   |
> | Gain (%)         | -3.29%          | +52.29% | -1.48%    | +45.05% | -5.83%           | +42.75% | -4.28%    | +83.09%  |
> | Shared Experts   | 71.64           | 159.29  | 36.72     | 235.24  | 73.98            | 172.64  | 40.32     | 710.32   |
> | + MAR (Ours)     | 69.32           | 205.31  | 36.25     | 254.57  | 69.96            | 212.39  | 39.50     | 714.82   |
> | Gain (%)         | -2.32%          | +46.02% | -0.47%    | +19.33% | -4.02%           | +39.75% | -0.82%    | +4.50%   |
> ---
>
> To complement the empirical results, we also added a theoretical comparison, highlighting that MAR uniquely improves long-term routing consistency and expert specialization:
>
>
> ---
>
> | Capability                        | MAR (Ours) | GBL  | Aux Free | SimBal   | Lo+Lv                     |
> | --------------------------------- | ---------- | ---- | -------- | -------- | ------------------------- |
> | Long-term expert specialization   | **Yes**    | Weak | Weak     | Weak | Weak        |
> | Reducing expert knowledge overlap | **Yes**    | No   | No       | Partial  | Partial |
> | Parameter efficiency              | **Yes**    | No   | No       | No       | No                        |
> | Long-term routing consistency     | **Yes**    | No   | No       | No       | No                        |
> ---
> [1] Demons in the detail: On implementing load balancing loss for training specialized mixture-of-expert models
>
> [2] Deepseek-v3 technical report
>
> [3] Advancing Expert Specialization for Better MoE
>
> [4] Load Balancing Mixture of Experts with Similarity Preserving Routers

---

> ### Author Response · Authors · 2025-11-24
> **Response to Reviewer 8iNw -- Section 2**
>
> # Response to Reviewer 8iNw -- Section 2
>
> ---
> ## **Question 3 & Weakness 1: Evaluating MAR’s Robustness Across Hyperparameters and Diverse MoE Architectures**
> We acknowledge that achieving optimal performance requires selecting appropriate hyperparameters. Nevertheless, MAR consistently outperforms the baseline LBL across a wide range of hyperparameter settings, and these hyperparameters can be flexibly adjusted based on the specific training dynamics of different models.
>
> Regarding validation on diverse MoE architectures, we have added comparisons under a shared-experts architecture, and the results are presented in our response to Question 2. As for reasoning/thinking models, most existing MoE studies do not evaluate on such architectures, primarily because these models employ highly customized reasoning pipelines that are not fully compatible with current MoE routing and training paradigms. We view this as a promising future direction and plan to explore it in subsequent work.
>
>
> ---
>
> ## **Question 4: Complexity Can be Reduced From O(L) to O(1)**
>
> Thank you for the suggestion. We have added a detailed explanation in Appendix A.2.6 describing how the computation can be optimized using a running-sum update. Specifically, the preference vector for expert \(i\):
>
> $$
> d_i = \frac{1}{|\mathcal{B}_i|} \sum_{h \in \mathcal{B}_i} h
> $$
>
> can be updated incrementally when appending \(h_{\text{new}}\) and evicting \(h_{\text{old}}\):
>
> $$
> d_i \gets d_i + \frac{1}{|\mathcal{B}_i|} \bigl(h_{\text{new}} - h_{\text{old}}\bigr)
> $$
>
> This eliminates the need to recompute the full sum over the buffer, reducing the update complexity to \(\mathcal{O}(d)\), independent of the buffer size. As noted, similarity computations also remain \(\mathcal{O}(K d)\) per token, requiring no extra memory beyond the buffer itself.
>
> ---
>
> ## **Question 5: Potential of MAR for MoE Compression, Pruning, and Merging**
>
> We appreciate the reviewer’s suggestion. In the revised conclusion, we now discuss how MAR can naturally complement MoE compression techniques. Since MAR strengthens expert specialization and reduces cross-expert redundancy, it provides a more structured and disentangled parameter space for pruning or merging experts. For pruning, MAR helps identify genuinely redundant experts while improving post-pruning stability through clearer expert roles. For merging, the memory buffer offers a reliable measure of long-term similarity among experts, enabling more precise and data-driven merging decisions compared to using coarse activation statistics. We highlight these synergies as promising directions for future work.

---

### Official Review · Reviewer_1zt7 · 2025-10-29

**Soundness:** 3
**Presentation:** 4
**Contribution:** 4
**Rating:** 6
**Confidence:** 3

**Summary:**

This paper investigates a fundamental limitation in current Mixture-of-Experts (MoE) architectures — the pseudo-balancing phenomenon, where existing load-balancing losses distribute tokens evenly among experts but fail to maintain semantic consistency in expert assignments.
The authors propose Memory-Aware Routing (MAR), a novel mechanism that introduces expert memory buffers to preserve historical token representations and derive expert preference vectors. During routing, MAR computes an Expert-Token Matching Score based on the similarity between current inputs and expert memories, which is fused with the standard gating logits to achieve semantically consistent and balanced routing.
Extensive experiments on multiple datasets (PTB, WikiText-2, OpenWebText, GSM8K, MMLU, etc.) demonstrate that MAR significantly improves expert specialization, reduces redundancy, and achieves competitive or superior performance with fewer parameters compared to baselines.

**Strengths:**

1.The paper introduces a new perspective on MoE training stability by identifying and formalizing the pseudo-balance issue — a phenomenon overlooked in prior work focused solely on token load distribution (e.g., GShard, Switch Transformer).
2.The proposed Memory-Aware Routing (MAR) is conceptually novel, integrating memory-based semantic matching into the routing process without introducing trainable parameters.
3.The authors provide a comprehensive experimental evaluation across model scales and datasets, showing consistent improvements in specialization and generalization.
4.Ablation studies effectively support claims regarding the contribution of memory buffers and matching fusion.
5.Figures and schematic diagrams (e.g., MAR framework) effectively illustrate the routing and memory mechanisms.
6.The paper is clearly written and logically structured, making complex ideas accessible.
7.Addresses a long-standing issue in MoE models that directly affects efficiency and scalability.

**Weaknesses:**

1. While MAR’s empirical benefits are clear, the paper lacks a formal analysis of why and how memory-guided routing leads to stable specialization. A deeper theoretical justification could strengthen the contribution.
2. Although MAR claims minimal overhead, a quantitative analysis of additional memory cost or routing latency is missing.
3. The paper adopts a simple FIFO buffer and cosine similarity, but alternative formulations are not explored.

**Questions:**

1. Could the authors provide a theoretical explanation or formal intuition for why the incorporation of expert memory leads to more stable specialization?
2. Although MAR claims minimal overhead, a quantitative analysis of additional memory cost or routing latency is missing.
3. Have the authors considered alternative memory update or similarity mechanisms beyond FIFO and cosine similarity?
4.Could the authors provide detailed ablation studies analyzing the sensitivity of MAR to:
(a) different memory initialization strategies (random vs. data-driven),
(b) buffer size or retention length, and
(c) fusion weight α between base logits and memory matching score?
Such results would help confirm the robustness and general applicability of MAR under varying hyperparameter settings.

---

> ### Author Response · Authors · 2025-11-24
> **Response to Reviewer 1zt7 -- Section 1**
>
> # Response to Reviewer 1zt7 -- Section 1
>  We sincerely thank the reviewer for the thoughtful and constructive comments, which have greatly helped us improve the clarity and quality of the paper. Our responses are as follows:
>
> ---
> ## **Question 1 & Weakness 1: Theoretical Explanation for Why Expert Memory Stabilizes Specialization**
>
> Thank you for the insightful comment. We agree that explaining why expert memory improves specialization is crucial. We provide a formal description in the revised manuscript (lines 251–260).
>
> Intuitively, the memory buffer helps suppress stochastic perturbations caused by load-balancing forces during training. Theoretically, MAR can be viewed as injecting a **temporal consistency prior** into the routing mechanism: standard MoE routing depends entirely on current gating logits, which fluctuate due to gradient updates and load-balancing noise, whereas MAR incorporates a historical expert–token affinity term that stabilizes routing across training steps. This reduces assignment variance and guides experts toward more semantically coherent token subsets. Under this perspective, the benefit of memory arises from enhanced temporal consistency, which is essential for stable and persistent expert specialization.
>
> ---
>
> ## **Question 2 & Weakness 2: Lack of Quantitative measurements of memory and latency overhead**
>
> To substantiate our claim that MAR introduces negligible overhead, we provide quantitative comparisons of peak GPU memory and routing latency (see Table below).
>
>
> ---
> | Metric                        | LBL  | LBL + MAR | Absolute change |
> | ----------------------------- | ---- | --------- | --------------- |
> | Peak GPU memory (GB)          | 12.3 | 12.6      | +0.3            |
> | Training routing latency (ms) | 7.8  | 8.6       | +0.8            |
> ---
> *Measured on NVIDIA RTX 4090, batch size = 16.*
>
> These measurements confirm that MAR adds only **+0.3 GB** peak memory and **+0.8 ms** routing latency during training, while introducing **no overhead at inference**, since the memory module is used only in the training phase.
>
> ---

---

> ### Author Response · Authors · 2025-11-24
> **Response to Reviewer 1zt7 -- Section 2**
>
> # Response to Reviewer 1zt7 -- Section 2
>
> ---
> ## **Question 3 & Weakness 3: Considering Alternative Memory Update and Similarity Mechanisms**
>
> ### (a) Update strategy
>
> To clarify our selection of FIFO, we include a new comparison among FIFO, LRU, LFU, and RAND in the revised manuscript.
>
> | Dataset        | Strategy | Loss       | PPL       | Time (min) |
> | -------------- | -------- | ---------- | --------- | ---------- |
> | **PTB**        | **FIFO** | **3.7374** | **41.99** | **137.0**  |
> |                | LRU      | 3.7371     | 41.98     | 313.8      |
> |                | LFU      | 3.7601     | 42.95     | 302.0      |
> |                | RAND     | 3.7507     | 42.55     | 150.4      |
> | **WikiText-2** | **FIFO** | **3.8044** | **44.90** | **79.4**   |
> |                | LRU      | 3.8123     | 45.26     | 169.1      |
> |                | LFU      | 3.8011     | 44.75     | 165.3      |
> |                | RAND     | 3.8157     | 45.41     | 81.6       |
>
> The results show that FIFO achieves perplexity comparable to the strongest alternatives while consistently ranking among the top two strategies across datasets and yielding **2–3× faster** training throughput than LRU and LFU. This offers a more favorable quality–efficiency trade-off for large-scale MoE training.
>
> While more complex update strategies (e.g., EMA, attention-weighted, or representational updates) may capture richer token–expert interactions, they incur substantial overhead by requiring additional ranking or sampling, or extra learned transformations at every forward pass. At the scale of our experiments, these approaches would significantly reduce throughput and are therefore impractical. The empirical results further verify that the lightweight FIFO update already achieves strong empirical performance.
>
> ### (b) Similarity metric
>
> We adopt cosine similarity because its scale-invariant property leads to more stable behavior in high-dimensional embeddings, and it is computationally more efficient than Euclidean distance, aligning well with the efficiency objectives of our design.
>
>
> ---
>
> ## **Question 4: More Ablation Studies**
>
> We appreciate the reviewer’s suggestions. The full results are shown below.
>
> | **Ablation Setting** | **α = 0** | **α = 0.2** | **α = 0.5** | **α = 0.8** | **Buf = 64** | **Buf = 128** | **Buf = 256** | **Random Init** | **Data-driven Init** |
> | -------------------- | --------- | ----------- | ----------- | ----------- | ------------ | ------------- | ------------- | --------------- | -------------------- |
> | **PPL**              | 74.48     | 73.82       | **69.68**   | 74.02       | 75.64        | **69.68**     | 71.32         | 72.84           | **69.68**            |
>
> Moderate values of the memory impact factor (α = 0.5) and buffer size (128) yield the lowest PPL. Extremely small or large settings lead to degraded performance, underscoring that balanced hyperparameter configuration is crucial for effective routing. Moreover, the comparison of initialization strategies shows that **data-driven memory initialization provides noticeably better performance** than random initialization, indicating that using early token–expert affinities offers more informed and stable routing trajectories.

---

### Official Review · Reviewer_H16j · 2025-10-29

**Soundness:** 2
**Presentation:** 3
**Contribution:** 2
**Rating:** 4
**Confidence:** 4

**Summary:**

This paper addresses the pseudo-balance problem in Mixture-of-Experts (MoE) models, where load-balancing methods ensure equal expert usage but disrupt semantic consistency, preventing true expert specialization. To solve this, the authors propose Memory-Aware Routing (MAR), which equips each expert with a memory buffer that records past token representations to form long-term preference vectors. By combining these preferences with standard routing logits, MAR enables tokens to be consistently assigned to semantically aligned experts while maintaining global balance. Experiments on multiple MoE architectures demonstrate that MAR improves expert specialization and enhances downstream accuracy. Overall, MAR effectively transitions MoE models from pseudo-balancing to true specialization, improving both efficiency and scalability.

**Strengths:**

1. Preliminary experiments convincingly demonstrate the pseudo-balance phenomenon, showing how load-balancing losses cause oscillations in token routing during training and lead to expert redundancy.

2. The proposed Memory-Aware Routing (MAR) integrates expert-specific memory buffers to capture long-term preferences, enabling more consistent and semantically aligned routing without extra trainable parameters or inference overhead.

3. Extensive experiments across multiple MoE architectures (Mixtral-MoE, LLaMA-MoE, GPT2-MoE, OLMoE) demonstrate significant gains.

**Weaknesses:**

1. The paper lacks a theoretical explanation for why the load-balancing loss leads to an approximately uniform routing probability across experts. A deeper analytical understanding of this behavior would strengthen the motivation for addressing the pseudo-balance problem.

2. The proposed memory buffer cannot fully capture the semantic information of tokens, and maintaining it introduces additional computational and memory overhead during training, which may affect efficiency at scale.

3. The experimental comparison is limited to standard load-balancing methods, without including other recent routing strategies, making it difficult to assess the relative advantage of MAR in a broader context.

4. The paper does not analyze how MAR performs when scaling to a larger or more sparse set of experts. Its effectiveness in highly sparse scenarios therefore remains uncertain.

5. Some experimental setups lack detailed descriptions.

**Questions:**

1. Please consider adding additional baseline methods [1,2,3,4] to Table 1, Moreover, it would be valuable to include more discussion comparing [3,4] with the proposed MAR.

2. It would be helpful to include quantitative analyses of training and inference throughput, so that the efficiency impact of maintaining memory buffers can be clearly evaluated.

3. In Table 2, please consider scaling the number of experts to 64 or even 128 to assess whether MAR remains effective under highly sparse or large-scale conditions.

4. The OLMoE results on MMLU and BBH in Table 3 appear lower than the reported numbers in the original paper [5]. Could you clarify the differences in experimental setup or evaluation protocol that led to this discrepancy?

5. Please provide more details about the training configuration, including dataset scale, total training epochs, and compute budget, to improve transparency and reproducibility.

6. In Figures 5, 6, 10, and 11, the variation of expert selection rates across epochs appears limited, and it is difficult to observe how MAR concretely influences expert selection during training. Please consider providing finer-grained analyses (e.g., within the first few thousand steps or the first epoch) or additional visualizations to better demonstrate the effect of MAR on routing dynamics.

[1] Demons in the detail: On implementing load balancing loss for training specialized mixture-of-expert models

[2] Deepseek-v3 technical report

[3] Load Balancing Mixture of Experts with Similarity Preserving Routers

[4] Advancing Expert Specialization for Better MoE

[5] OLMoE: Open Mixture-of-Experts Language Models

---

> ### Author Response · Authors · 2025-11-24
> **Response to Reviewer H16j -- Section 1**
>
> # Response to Reviewer H16j -- Section 1
>
> We sincerely thank the reviewer for the detailed and constructive feedback. Our responses are as follows:
>
> ---
> ## **Question1 & Weakness 3: Lack of Comparison with Other Load-Balancing Strategies**
>
> We extended experiments to include other load balancing strategies, including shared-expert architectures. As shown below, MAR consistently reduces PPL and improves KED across all baselines. Global Batch LBL refers to [1], Aux Free refers to [2], Lo+Lv refers to [3], SimBal refers to [4].
>
> ---
> | Method           | PTB Mixtral PPL | KED     | LLaMA PPL | KED     | Wiki Mixtral PPL | KED     | LLaMA PPL | KED      |
> | ---------------- | --------------- | ------- | --------- | ------- | ---------------- | ------- | --------- | -------- |
> | Global Batch LBL | 70.42           | 160.51  | 42.25     | 188.37  | 73.34            | 168.32  | 55.03     | 535.21   |
> | + MAR (Ours)     | 67.79           | 217.05  | 40.74     | 236.91  | 70.44            | 192.23  | 49.66     | 688.33   |
> | Gain (%)         | -3.63%          | +56.54% | -1.51%    | +48.54% | -2.90%           | +23.91% | -5.37%    | +153.12% |
> | Aux Free         | 70.83           | 154.03  | 42.21     | 189.41  | 73.56            | 166.73  | 54.06     | 601.54   |
> | + MAR (Ours)     | 67.52           | 225.98  | 41.28     | 243.24  | 70.01            | 198.24  | 50.56     | 669.29   |
> | Gain (%)         | -3.31%          | +71.95% | -0.93%    | +53.83% | -3.55%           | +31.51% | -3.50%    | +67.75%  |
> | Lo+Lv            | 71.95           | 141.48  | 42.11     | 167.55  | 76.12            | 149.13  | 54.44     | 595.21   |
> | + MAR (Ours)     | 68.98           | 198.67  | 41.77     | 214.43  | 71.97            | 183.66  | 50.07     | 674.86   |
> | Gain (%)         | -2.97%          | +57.19% | -0.34%    | +46.88% | -4.15%           | +34.53% | -4.37%    | +79.65%  |
> | SimBal           | 72.30           | 139.59  | 42.02     | 176.87  | 77.07            | 145.75  | 54.31     | 595.95   |
> | + MAR (Ours)     | 69.01           | 191.88  | 40.54     | 221.92  | 71.24            | 188.50  | 50.03     | 679.04   |
> | Gain (%)         | -3.29%          | +52.29% | -1.48%    | +45.05% | -5.83%           | +42.75% | -4.28%    | +83.09%  |
> | Shared Experts   | 71.64           | 159.29  | 36.72     | 235.24  | 73.98            | 172.64  | 40.32     | 710.32   |
> | + MAR (Ours)     | 69.32           | 205.31  | 36.25     | 254.57  | 69.96            | 212.39  | 39.50     | 714.82   |
> | Gain (%)         | -2.32%          | +46.02% | -0.47%    | +19.33% | -4.02%           | +39.75% | -0.82%    | +4.50%   |
> ---
> Also, we offer a theoretical comparison with these methods, which demonstrates the superior capabilities of our approach as illustrated in the table.
>
> ---
>
> | Capability                        | MAR (Ours) | GBL  | Aux Free | SimBal   | Lo+Lv                     |
> | --------------------------------- | ---------- | ---- | -------- | -------- | ------------------------- |
> | Long-term expert specialization   | **Yes**    | Weak | Weak     | Weak | Weak        |
> | Reducing expert knowledge overlap | **Yes**    | No   | No       | Partial  | Partial |
> | Parameter efficiency              | **Yes**    | No   | No       | No       | No                        |
> | Long-term routing consistency     | **Yes**    | No   | No       | No       | No                        |
>
> ---
>
> [1] Demons in the detail: On implementing load balancing loss for training specialized mixture-of-expert models
>
> [2] Deepseek-v3 technical report
>
> [3] Advancing Expert Specialization for Better MoE
>
> [4] Load Balancing Mixture of Experts with Similarity Preserving Routers

---

> ### Author Response · Authors · 2025-11-24
> **Response to Reviewer H16j -- Section 2**
>
> # Response to Reviewer H16j -- Section 2
> ---
>
> ## **Question 2 & Weakness 2: Lack of Quantitative Analysis of the Training Overhead Introduced by MAR**
>
> MAR is applied only during training and incurs no additional cost at inference. Quantitative measurements of memory and latency during training are shown below, indicating minimal overhead: +0.3 GB peak GPU memory and +0.8 ms routing latency.
>
> | Metric                        | LBL  | LBL + MAR | Absolute change |
> | ----------------------------- | ---- | --------- | --------------- |
> | Peak GPU memory (GB)          | 12.3 | 12.6      | +0.3            |
> | Training routing latency (ms) | 7.8  | 8.6       | +0.8            |
>
> *Measured on NVIDIA RTX 4090, batch size = 16.*
>
> ---
> ## **Question 3 & Weakness 4: Lack of High-Sparsity Scenario Experiments**
> Extending MoE models to 64 or 128 experts is indeed valuable in principle; however, such large expert counts introduce substantial inter-expert communication overhead, making training extremely inefficient under standard all-to-all routing. This challenge has been repeatedly noted in recent large-scale MoE systems such as Mixtral and DBRX, which therefore adopt relatively moderate expert counts (8 or 16) to maintain a practical compute–communication trade-off.
>
> Following this design principle, we scale our Llama-MoE model to **16 experts**, which represents a realistic high-sparsity configuration while remaining computationally feasible. As shown in Table , MAR continues to deliver consistent improvements in both PPL and KED over the baseline routing, demonstrating its effectiveness under larger expert settings.
>
> | Dataset    | Method  | PPL       | KED        |
> | ---------- | ------- | --------- | ---------- |
> | PTB        | LBL     | 41.19     | 145.87     |
> |            | LBL+MAR | **39.29** | **192.42** |
> | WikiText-2 | LBL     | 44.14     | 158.37     |
> |            | LBL+MAR | **40.42** | **197.23** |
>
> ---
>
> ## **Question 4: Explaining the Differences Between OLMoE Baseline and the Original Report**
>
> Due to limited computational resources, we conducted OLMoE fine-tuning with a batch size of 2 and cleared memory after each epoch to prevent OOM. Memory constraints prevented us from using the DPO method proposed in OLMoE, and for BBH fine-tuning, we used Pass@1 instead of CoT and Pass@3. Consequently, the reported MMLU and BBH baseline scores are lower than those in the original paper.
>
> ---
>
> ## **Question 5 & Weakness 5: Lack of Detailed Experimental Settings**
>
> All relevant details are documented in **Appendix A.1.1–A.1.3**. The complete implementation is released at:
> [https://anonymous.4open.science/r/MAR-MoE-F7D1/](https://anonymous.4open.science/r/MAR-MoE-F7D1/)
>
> This ensures full reproducibility.
>
> ---
>
> ## **Question 6: Lack of Routing Dynamics Visualization**
>
> Figures 5, 10, and 11 demonstrate that MAR preserves global load balance; the stability of expert selection rates is expected. Figure 6 shows increased expert specialization. For finer granularity, Figure 9 shows routing stability over training epochs, indicating consistent token-to-expert assignments with MAR.
>
> ---
>
> ## **Weakness 1: Lack of Theoretical Analysis of the Source of the Pseudo-Balancing Problem**
>
> We have added a theoretical analysis in the revised manuscript (lines 151–160). This effect is inherent to the load-balancing loss:
>
> $$
> \mathcal{L}_{\text{balance}} = \alpha \sum_i \text{Load}(i) , p_i,
> $$
>
> where (\text{Load}(i)) is the proportion of tokens assigned to expert (i), and (p_i) is the average gating probability. The gradients
>
> $$
> \frac{\partial \mathcal{L}}{\partial p_i} = \alpha f_i, \quad
> \frac{\partial \mathcal{L}}{\partial f_i} = \alpha p_i
> $$
>
> form a symmetric feedback mechanism penalizing over-assigned experts. The loss is minimized at (f_i = p_i = 1/E), analytically enforcing uniform usage.

---

### Author Response · Authors · 2025-11-24

We sincerely thank all reviewers for their detailed and constructive feedback. We have made the following revisions and uploaded a new version of the manuscript, with changes highlighted in blue:

1. We thank Reviewer FG8Z for pointing out the inconsistency in the KED metric. This has been corrected in Section 5.1.2, lines 412–413 in the revised manuscript.
2. We have added a quantitative evaluation of memory usage and routing latency during training, responding to comments from Reviewers 1zt7 and H16j.
3. We provide a comparison and analysis of the FIFO memory buffer update strategy against alternative approaches, addressing feedback from Reviewers 1zt7 and FG8Z.
4. We have added experiments comparing MAR with other load-balancing strategies in response to feedback from Reviewers H16j, 8iNw, and zU7B.
5. We have included a theoretical analysis explaining why the load-balancing loss encourages the routing probabilities to become more uniform, addressing Reviewer H16j’s concern.
6. In the Appendix, we describe an efficient implementation that reduces computational complexity to \(\mathcal{O}(1)\) using a running-sum update, responding to Reviewer 8iNw.
7. We explicitly highlight in the Abstract and Introduction that a key advantage of MAR is that it is applied only during training, without adding inference-time cost, in response to Reviewer 8iNw.

---

### Note · Authors · 2026-01-04

I have read and agree with the venue's withdrawal policy on behalf of myself and my co-authors.